# Deep terrestrial indigenous microbial community dominated by *Candidatus* Frackibacter
Sian E. Ford [1], Greg F. Slater [1 ✉], Katja Engel[2], Oliver Warr [3,4], Garnet S. Lollar[3], Allyson Brady[1,5], Josh D. Neufeld [3] & Barbara Sherwood Lollar[2,6]

Characterizing deep subsurface microbial communities informs our understanding of Earth's biogeochemistry as well as the search for life beyond the Earth. Here we characterized microbial communities within the Kidd Creek Observatory subsurface fracture water system with mean residence times of hundreds of millions to over one billion years. 16S rRNA analysis revealed that biosamplers well isolated from the mine environment were dominated by a putatively anaerobic and halophilic bacterial species from the *Halobacteroidaceae* family, *Candidatus* Frackibacter. Contrastingly, biosamplers and biofilms exposed to the mine environment contained aerobic *Sphingomonas* taxa. $\delta^{13}C$ values of phospholipid fatty acids and putative functional predictions derived from 16S rRNA gene profiles, imply *Candidatus* Frackibacter may use carbon derived from ancient carbon-rich layers common in these systems. These results indicate that *Candidatus* Frackibacter is not unique to hydraulically fracked sedimentary basins but rather may be indigenous to a wide range of deep, saline groundwaters hosted in carbon-rich rocks.

The deep terrestrial subsurface of Earth is known to harbor microbial life at depths of up to several kilometers[1–5]. Microorganisms that inhabit such isolated biomes do so under conditions that are commonly far removed from (near) surface conditions, including increased temperature, pressure, and salinity, pH extremes, and/or nutrient limitations. Despite these challenges, many deep subsurface ecosystems contain a diverse array of microbial communities[4,6–12]. In other, often deeper, subsurface biomes, especially those located in ancient groundwater systems isolated from the surface on long time scales of millions to billions of years[13], the ecosystems have been shown to be dominated by only one or two phylotypes[3,6,14–18]. In all systems, the host rock composition and the availability of dissolved compounds in groundwaters influence which organisms and metabolisms are supported[4,11,19–21], but the details of these controls, and further characterization of the nature and distribution of community biodiversity, remain a major focus of investigation of deep surface microbiology research.

In some terrestrial analogue environments, chemolithotrophic subsurface biomes can exist independently of the photosphere and atmosphere and can be self-reliant for sources of organic carbon and energy[3,5,16,22–25]. Understanding origins, metabolic activities, and biosignatures of deep subsurface communities, and, in particular, determining the extent to which

there are indigenous populations in these systems, also provides context to understand potential subsurface communities elsewhere, such as on Mars, Enceladus, or Europa[23,26–38]. The Kidd Creek fracture fluids have been cited as a potential analogue for ocean world exploration due to their geochemical properties and long-term isolation from the surface and its processes[36–38]. With the recent detection of the near-surface presence of organic compounds in Gale Crater[26], interest in the potential habitability of the Martian subsurface has been reignited. Although there is no permanent presence of liquid water on the surface of Mars, the present-day Martian subsurface could contain saline groundwaters in deep fractures like those found in the Earth's terrestrial crust[23,27–31]. Over deep time, fracture fluids typically increase in salinity due to long-term water-rock interactions[32,33]. Fluids isolated within the Martian subsurface may similarly contain highly saline fluids or brines, bringing into the forefront the importance of halophilic life-strategies when considering long-term habitability both on Earth and beyond[29,33–37].

Studying microbial communities within the Earth's deep continental subsurface is challenging due to the high cost of drilling to depth and the typical low biomass of the resident microbial communities[39–42]. However, more widespread access to the continental subsurface can be gained via deep

[1]School of Earth, Environment and Society, McMaster University, Hamilton, ON, Canada. [2]University of Waterloo, Waterloo, ON, Canada. [3]Department of Earth Sciences, University of Toronto, Toronto, ON, Canada. [4]Department of Earth and Environmental Sciences, University of Ottawa, Advanced Research Complex, Ottawa, ON, Canada. [5]Department of Biology, Carlton University, Ottawa, ON, Canada. [6]Institut de Physique du Globe de Paris (IPGP) Université Paris Cité, Paris, France. ✉e-mail: gslater@mcmaster.ca

subsurface laboratories and active mines around the world. In some cases, dedicated subsurface science observatories, such as the Kidd Creek Observatory (KCO) on the Canadian Shield, can allow for long-term monitoring and collection of large volume samples in such settings[24,43]. Evaluating the impact of mining activities on in situ communities is a critical component of methodology, including the potential introduction of microbial contamination via transportation or water utilization/dewatering activities within the mine [[43], and references therein]. Hence, in addition to collecting in situ samples, it is also critical to evaluate potential contamination sources through multiple approaches, including water and rock geochemistry, experimental replication, and collection of control samples representative of mine influences, such as service water (SW) involved in mine operations and typically sourced from surface lakes) and air exposed surfaces[43–45]. Using such approaches, previous studies demonstrated cycling of microbial carbon sources by indigenous subsurface communities and revealed the metabolic capability and interactions of indigenous microorganisms dwelling in deep subsurface fracture fluids [e.g. [20,24][,33,43,45–58]]. Through both cultivation and non-cultured based genomic approaches, two previous studies established the presence of indigenous microbiota at Kidd Creek that was dominated by sulfate reducers[43,45]. However, the full role, range, and extent of microbial cycling of potential electron acceptors and donors remained unresolved in previous studies.

The KCO is located at 2.4 km below surface within an active mine in the 2.7-billion-year-old rock of the Canadian Shield near Timmins (Ontario, Canada) and is located on the traditional territories of the Anishinabewaki and Cree. This location has been the focus of subsurface biosphere studies since the 1990s[46] and was established as an international

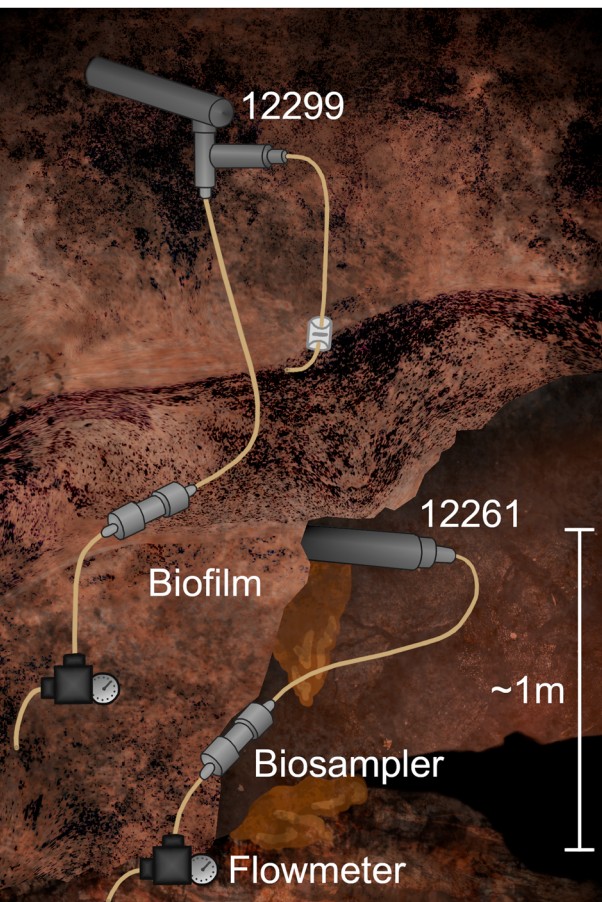

**Fig. 1 | Schematic of the Kidd Creek Observatory at 7850 ft below the surface.** Boreholes 12299 and 12261 are illustrated as well as the biofilm formed around and below 12261. The schematic shows the approximate configuration of the sampling apparatus.

subsurface science observatory in 2008. The environment monitored and sampled at this observatory has previously been established as a potential analogue for subsurface microbial systems that might exist on Mars or other solar system bodies[23,24,43,47–49]. The site geology has been described previously, most recently by Li and colleagues[50]. In brief, the Kidd Creek deposit is a 2.4 billion-year-old massive sulfide deposit located within the Abitibi Greenstone belt of the Superior Province of the Canadian Shield. Rhyolitic layers are bordered by high silica rhyolite flows and tholeiitic basalts, and up to 500 m of carbonate-altered komatiitic flows[51]. In a 2013 noble gas study, the average mean residence time (MRT) for the fracture waters was between 1.1 to 1.7 Ga[48], with subsequent studies estimating the youngest fluid components contributing to the overall mean groundwater system to be in the range of 150–650 Ma[13,43,52]. Accompanying the fracture waters is substantial gas flow, which includes abiogenic methane ($CH_4$), higher hydrocarbons (ethane, propane, butane)[53–57], and molecular hydrogen ($H_2$) produced by a combination of radiolysis and hydration of local mafic and ultramafic rocks (serpentinization)[58,59]. Li et al.[23] found that in addition to $H_2$ production, indirect radiolytic oxidation of sulfide minerals produce a long-term source of dissolved sulfate in the fracture fluids. Most recently, Sherwood Lollar et al.[24] evaluated the potential role of radiolytic processes in the abiotic production of acetate and formate, contributing to the dissolved organic carbon pool in the fracture waters.

Recent work within the Kidd Creek system focused on habitability and used the most probable number (MPN) cultivation methodology to investigate anaerobic microorganisms and their metabolisms in several of the same boreholes that are the focus of this study[43]. In this previous work, MPN studies indicated that the fracture fluids contained low abundance indigenous microbial communities ($\sim10^3$–$10^4$ cells/mL), which could perform both autotrophic and alkane-oxidizing sulfate reduction. The indigenous nature of these microorganisms was determined based on the significant differences in biomass and dominant metabolisms (as inferred from MPN cultures) compared to the service water samples[43]. Further genetic analysis was performed on several boreholes at the same observatory, including one that had been sealed by a packer from the surrounding mine environment for seven years at the time of sampling[45]. The microbial community detected in this borehole (FW12299) was comprised of moderately halophilic bacteria (e.g., *Chromohalobacter* and *Marinobacter*) as well as homoacetogens from the genus *Fuchsiella*. The presence of such bacteria in a deep subsurface system is expected, as recent work suggests that microorganisms that are isolated on long time scales can potentially be evolutionarily static in these settings relative to model organisms used in evolutionary studies[60]. The fracture waters in the KCO, may have been isolated for hundreds of millions of years to as many as a few billion years[31,43,52]. While any life may be as young as the "youngest drop" in the fracture water system, currently estimated at 150 Ma, these systems are some of the most temporally isolated subsurface biomes yet investigated. Genomic and phospholipid investigations have not been carried out to date.

The goal of the current study was to further investigate the indigenous microbial communities living within the anoxic hypersaline fracture fluids of the deep terrestrial subsurface at Kidd Creek. This was achieved through the long-term instrumentation of multiple flowing boreholes in the KCO at a depth of 7850 ft (2.39 km) below surface level. Monitoring of the fracture waters provided a well-characterized geochemical and hydrogeologic context within which to employ multiple complementary methods on large-volume sample. A combination of phospholipid, stable isotope, and genetic techniques were used to characterize the microbial communities at the site and to investigate the possible metabolic strategies they may employ. To achieve this, fracture water (FW) was passively flowed through pre-sterilized low-carbon stainless steel biosampler units at the outflow of two of the Observatory boreholes, 12261 and 12299 for up to 8 months (Fig. 1). The material collected by the biosamplers along with surface biofilms from the adjacent mine walls, and the service water used by the mine, were analyzed for microbial community abundance and diversity using phospholipid fatty acids (PLFA) and 16S rRNA gene sequencing. To confirm the indigenous origin of detected microorganisms, sample results were compared with

results from potential sources of contamination from the external mine environment and service water.

## Results

### Geochemical context of Kidd Creek Fracture Waters

Geochemistry, dissolved inorganic carbon (DIC) ($\delta^{13}C_{DIC}$), and $CH_4$ ($\delta^{13}C_{CH4}$), as well as stable oxygen isotopes ($\delta^{18}O_{H2O}$) and hydrogen isotopes ($\delta^2H_{H2O}$) of the fracture water (FW) and service water (SW), were measured for both boreholes included in this study (Supplementary Table 1). Data were largely consistent with geochemical results from earlier dates reported[23,24,31,43,61]. Fluid and gas flow rates were monitored, with gas flow rates ranging between 2729 mL/min to 5938 mL/min and fluid flow rates ranging from 127 mL/min to 280 mL/min during the period of the study (Supplementary Tables 1, 2). Continuous natural flushing of both boreholes is due to artesian flow which contributes to reduced contamination from mine activities[44]. Fluids from 12299 and 12261 were highly saline (3.6 M), have circumneutral pH (6.32–6.7), and moderate temperatures (23.6–25.0 °C) (Supplementary Table 1). The two boreholes, FW12261 and FW12299, were not statistically different (ANOVA, $p > 0.01$) in terms of geochemical species, temperature, $\delta^{13}C_{CH4}$, $\delta^{18}O_{H2O}$, and $\delta^2H_{H2O}$, and were geochemically stable over the sampling intervals. This is consistent with previous studies that have demonstrated that the two boreholes are likely draining the same fracture networks in the host rock[23,24,43,54,61]. The service water was geochemically and isotopically distinct from both FW12261 and FW12299 (ANOVA, $p < 0.01$), and had $\delta^{18}O_{H2O}$ and $\delta^2H_{H2O}$ values consistent with its origin from a surface lake as established by previous studies[23,31,43], while the isotopic composition of FW12299 and FW12261 confirm no evidence of mixing with that service water[43].

### PLFA abundance and isotopic compositions

The biosampler glass wool and biofilm samples each had total PLFA abundances of $10^5$–$10^6$ pmole (Table 1), reflecting both cells from the outflow water and microorganisms that may have grown on biosampler glass wool surfaces during the long installation period. The PLFA-associated biomass for biosamplers was consistent between sampling time points for a given fracture water and between fracture waters. Three biosampler process blanks yielded PLFA concentrations equivalent to ≤0.03% of the biosampler concentrations. The consistency of the PLFA concentrations detected indicated a similar magnitude of microbial biomass was collected in all biosamplers over the duration of their installation. PLFA profiles were generated for each sample, represented by the relative (mol%) abundance of PLFA classes (Fig. 2), as well as for all individual PLFAs (Supplementary Table 3). The $\delta^{13}C$ values were determined for PLFA that were sufficiently abundant for analysis (Table 2). Analysis using Spearman's rank correlation coefficient ($\rho$) revealed pair-wise sample similarity based on Spearman's $\rho$, the values for which are visualized as a heatmap (Fig. 3).

A total of 17 PLFA were identified in the 2016 biofilm, with four PLFA making up 84% of the total abundance (16:0, 18:0, 16:1$^9$, and 18:1c$^9$). The 2017 biofilm sample had 23 PLFA peaks with the same four PLFA making up 76% of the total abundance (Table 1, Supplementary Table 3). The $\delta^{13}C$-PLFA values for the biofilms were likewise consistent between the two sampling years. The biofilm sample from 2016 had an average $\delta^{13}C_{PLFA}$ of −12.2‰ (range: −13.7‰ to −10.4‰; $n_{PLFA} = 3$) and the biofilm from 2017 had an average $\delta^{13}C_{PLFA}$ of −13.0‰ (range: −14.6‰ to −11.3‰; $n_{PLFA} = 3$) (Table 2). The PLFA profiles of biofilm samples from the mine walls were moderately correlated to each other ($\rho = 0.44$, $p < 0.01$) and distinct from the service water (2016: $\rho = -0.08$, $p = 0.55$; 2017: $\rho = -0.15$,

---

**Table 1 | Characterization of phospholipid fatty acid abundance of the biomass detected in Kidd Creek**

| Sample | Sample size (g or mL) | Number of PLFA | PLFA abundance (pmol/g or pmol/mL) | Cellular abundance (cells/g or cells/mL)[a] |
|---|---|---|---|---|
| FW12261-2017A | 21.3 g | 25 | $1.5 \times 10^4$ | $2.9 \times 10^8$ |
| FW12261-2017B | 21.3 g | 21 | $8.0 \times 10^4$ | $1.6 \times 10^8$ |
| [a]FW12261-2018 | 21.3 g | 20 | $9.4 \times 10^4$ | $1.8 \times 10^8$ |
| FW12299-2017A | 21.3 g | 23 | $3.6 \times 10^4$ | $7.1 \times 10^8$ |
| FW12299-2017B | 21.3 g | 20 | $6.6 \times 10^4$ | $1.3 \times 10^8$ |
| FW12299-2018 | 21.3 g | 17 | $8.5 \times 10^4$ | $1.7 \times 10^8$ |
| Biofilm 2016 | 6.3 g | 17 | $1.1 \times 10^5$ | $7.3 \times 10^9$ |
| Biofilm 2017 | 9.8 g | 22 | $1.3 \times 10^5$ | $5.7 \times 10^9$ |
| Service Water 2018 | 17300 mL | 18 | $6.6 \times 10^6$ | $1.7 \times 10^6$ |

[a]Conversion factor of $2 \times 10^4$ cells/pmol$_{PLFA}$ (Green & Scow 2000).

---

**Fig. 2 | Relative abundance (mole %) of PLFA classes detected in each biosampler, biofilm, and the service water, pooled from all individual, quantifiable PLFA (Supplementary Table 3).** PLFA classes are grouped into: Saturates (black); Monounsaturates (dark grey); Polyunsaturates (light grey); Terminally branched (dark blue); Midbranched (light blue); Cyclopropyl (red) and Other (orange).

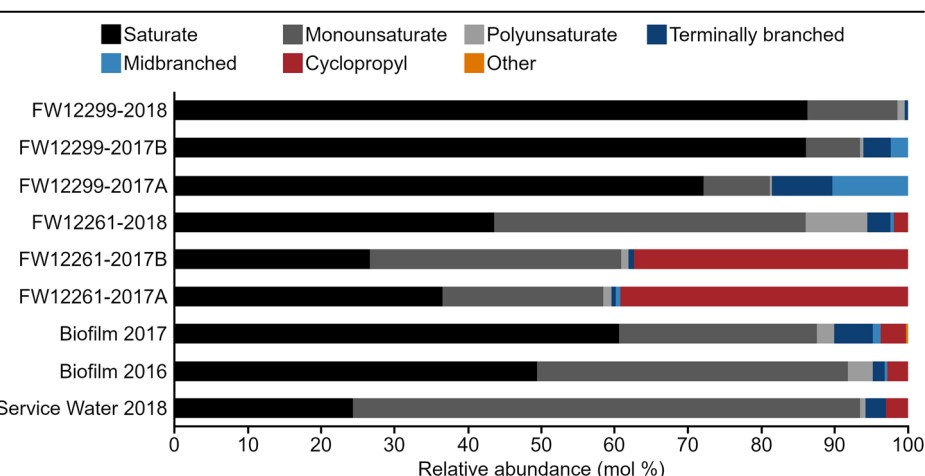

**Table 2 | The δ¹³C (‰) values of individual PLFA detected in Kidd Creek biosamplers, biofilms and service water. The average of these values is presented with standard deviation**

| PLFA | FW12261 2017A | FW12261 2017B | FW12261 2018 | FW12299 2017A | FW12299 2017B | FW12299 2018 | Biofilm 2016 | Biofilm 2017 | Service Water 2018 |
|---|---|---|---|---|---|---|---|---|---|
| 12:0 | | | −27.9 ± 0.1 | −28.8 ± 0.1 | −29.2 ± 0.4 | −30.3 ± 0.4 | | | |
| 14:0 | | | −25.4 ± 0.1 | | −28.1 ± 0.5 | −30.0 ± 0.5 | | | |
| 16:0 | −11.8 ± 0.3 | −12.1 ± 0.5 | −23.3 ± 0.1 | −29.9 ± 0.1 | −30.6 ± 0.4 | −29.7 ± 0.7 | −10.4 ± 0.02 | −11.3 ± 0.2 | −34.4 ± 0.2 |
| 18:0 | −18.9 ± 0.3 | −16.7 ± 0.5 | | −28.9 ± 0.3 | −29.3 ± 0.4 | −29.9 ± 0.6 | −12.4 ± 0.4 | −13.2 ± 0.3 | |
| i15:0 | | | −26.4 ± 0.3 | | | | | | |
| 16:1⁹ | −10.2 ± 0.8 | −13.5 ± 0.2 | −17.2 ± 0.2 | −24.0 ± 0.1 | −24.4 ± 0.2 | | | | −33.9 ± 0.3 |
| 18:1⁹c | | | | | | −27.4 ± 0.5 | −13.7 ± 0.1 | −14.6 ± 0.3 | |
| 18:1⁹t | | | −27.1 ± 0.3 | | −28.2 ± 2.3 | | | | −30.9 ± 0.5 |
| 18:1ˣ | −18.0 ± 0.2 | −16.0 ± 0.2 | | | −25.9 ± 0.6 | | | | |
| 18:2⁹,¹² | | | −29.2 ± 3.1 | | | | | | |
| cy17:0Δ⁹c | −12.6 ± 0.003 | −11.3 ± 0.5 | | | | | | | −39.0 ± 0.9 |
| cy19:0Δ⁹c | −16.0 ± 0.2 | −16.1 ± 0.02 | | | | | | | |
| Average | −14.6 ± 3.5 | −14.3 ± 2.3 | −24.9 ± 4.3 | −27.9 ± 2.7 | −27.9 ± 2.3 | −29.3 ± 1.2 | −12.2 ± 1.7 | −13.0 ± 1.7 | −34.6 ± 3.4 |

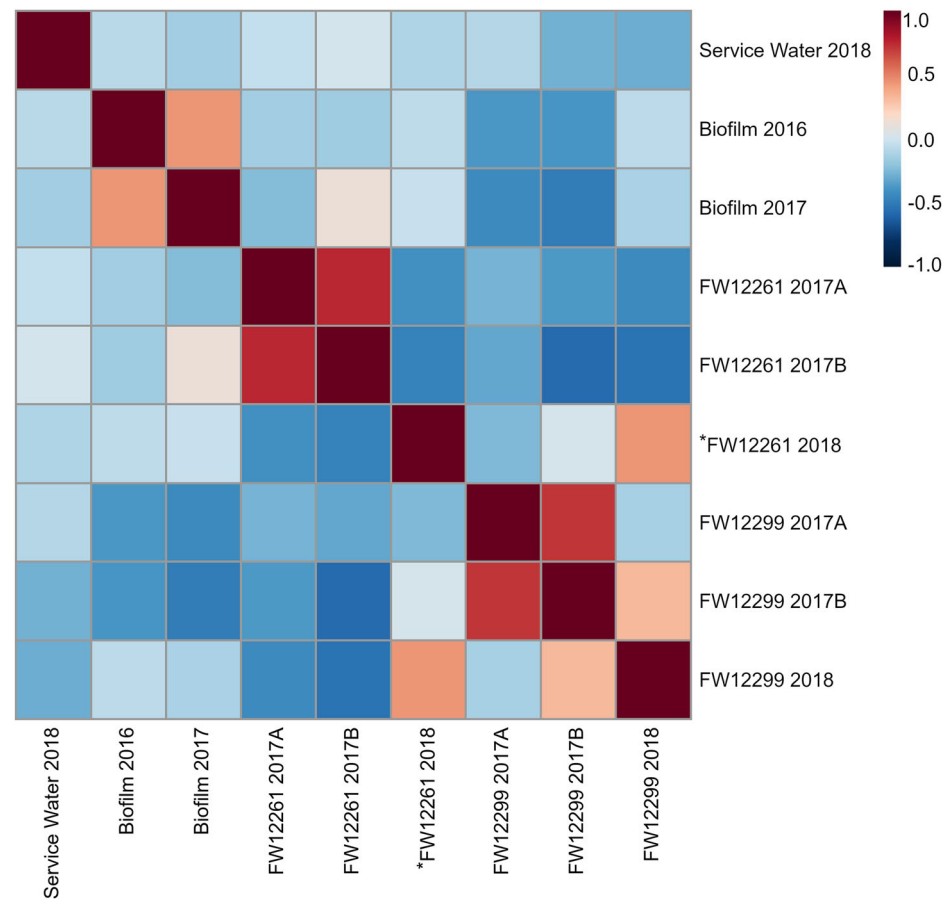

**Fig. 3 | Correlation heatmap for the PLFA profile of each biosampler, biofilm and the service water generated using Spearman's rank correlation (ρ). More positive numbers (red) indicate a strong, positive correlation.** Numbers closer to zero indicate no correlation. More negative numbers (dark blue) indicate a weak negative correlation. The star (*) designates the biosampler that became detached from the borehole during sampling (*FW12261-2018).

$p = 0.25$) and from the biosamplers (FW12261: $-0.13 \leq \rho \leq$ -0.04, $0.41 \leq p \leq 0.92$; FW12288: $-0.29 \leq \rho \leq$ -0.28; $0.02 \leq p \leq 0.03$) (Fig. 3).

Biosampler FW12261-2017A had 25 PLFA peaks, of which six made up 88% of the total abundance: 16:0, 18:0, 18:1c⁹, 18:1t⁹, cy17:0Δ⁹c, and cy19:0Δ⁹c (Table 1, Supplementary Table 3). The FW12261-2017B biosampler had 21 PLFA peaks with the same six PLFA representing 86% of the total PLFA abundance (Table 1, Supplementary Table 3). The average δ¹³C$_{PLFA}$ value for FW12261-2017A and 2017B was −14.4‰ (range: −17.8‰ to −11.9‰; $n_{PLFA} = 6$) (Table 2). As noted in the Methods, the

*FW12261-2018 biosampler detached at an unknown point prior to retrieval. This biosampler had 20 PLFA in total, with four PLFA representing 79% of the total abundance: 16:0, 18:0, 18:1⁹, 18:2tˣ·ʸ (Table 1, Supplemental Table S3). In addition, the δ¹³C$_{PLFA}$ values for *FW12261-2018 were much more depleted than the biosamplers of the previous year, with an average of −25.8‰ (range: −29.7‰ to −17.2‰; $n_{PLFA} = 8$) (Table 2). All FW12261 biosamplers were statistically different from the service water. Biosamplers FW12261-2017A and FW12261-2017B had PLFA profiles were strongly correlated with each other (ρ = 0.75, $p < 0.01$)

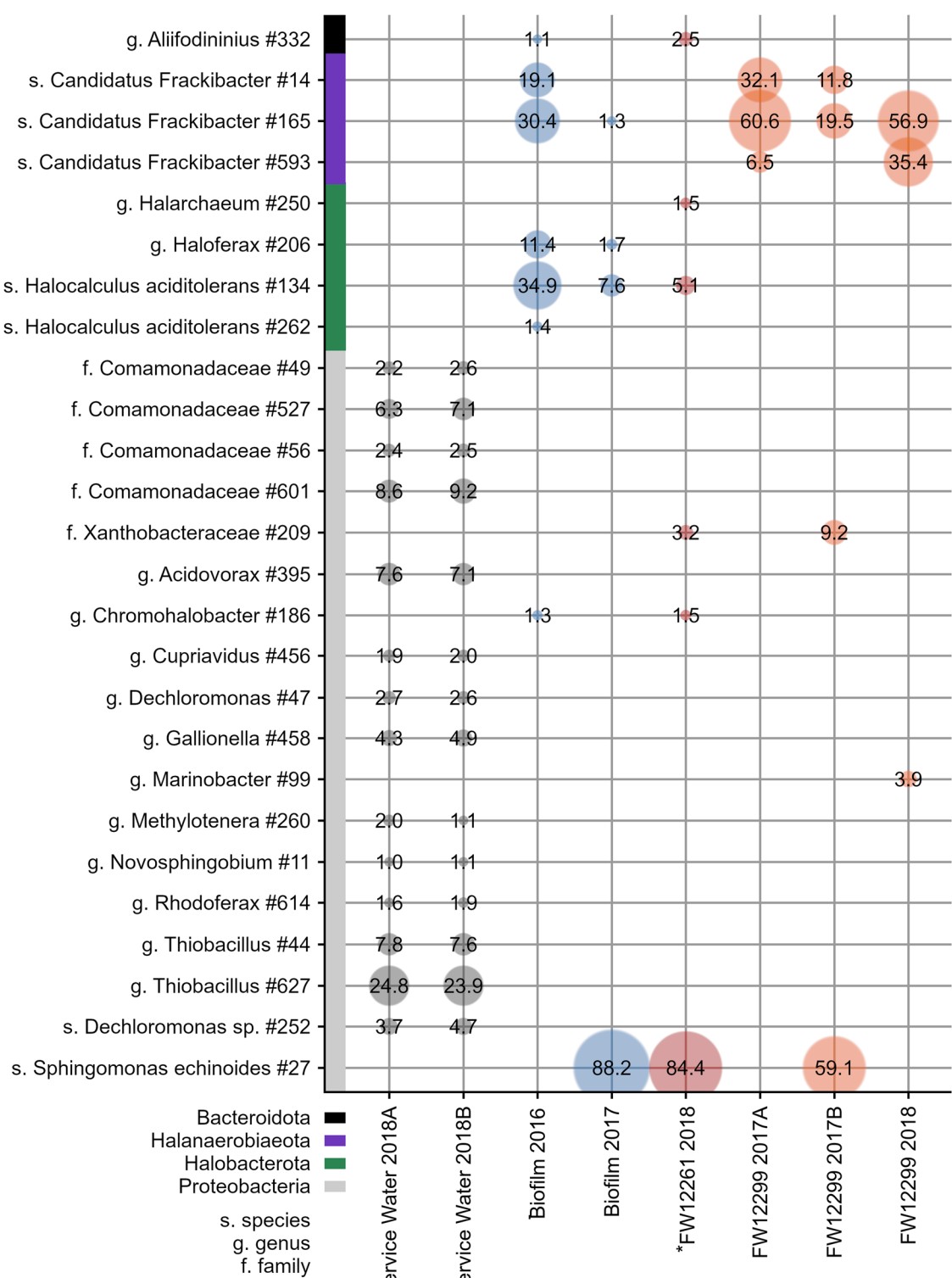

**Fig. 4 | 16S rRNA gene analysis of microbial communities detected in the service water, biofilms, and borehole biosamplers.** The relative abundance of each individual taxa at or above 1% relative abundance are shown. The sample designated with a star (*), was the biosampler that became detached from the borehole during sampling (*FW12261-2018). The service water sample from 2018 was analyzed in duplicate. Each ASV is followed by a randomly generated ASV number.

but were not similar to any other biosamplers (FW12261 2018: $\rho = -0.41$, $p < 0.01$ and $\rho \le -0.47$, $p < 0.01$; FW12288 2017A: $\rho = -0.27$, $p < 0.05$ and $\rho \le -0.32$, $p < 0.05$; FW12288 2017A: $\rho = -0.37$, $p < 0.01$ and $\rho \le -0.0058$, $p < 0.01$; FW12288 2018: $\rho = -0.43$, $p < 0.01$ and $\rho \le -0.0053$, $p < 0.01$;) (Fig. 3). The PLFA profile for the *FW12261-2018 biosampler was statistically different from the 2017 profiles, as well as from all other biosamplers,

excepting FW12299-2018 to which it was only moderately similar ($\rho = 0.44$, $p < 0.01$) (Fig. 3).

In the FW12299-2017A biosampler there were 23 PLFA, of which five made up 78% of the total abundance: 12:0, 16:0, 18:0, br15:0, and i15:0. The FW12299-2017B biosampler had 20 PLFA with only three saturated PLFA representing 78% of the total abundance: 12:0, 16:0, and 18:0 (Table 1,

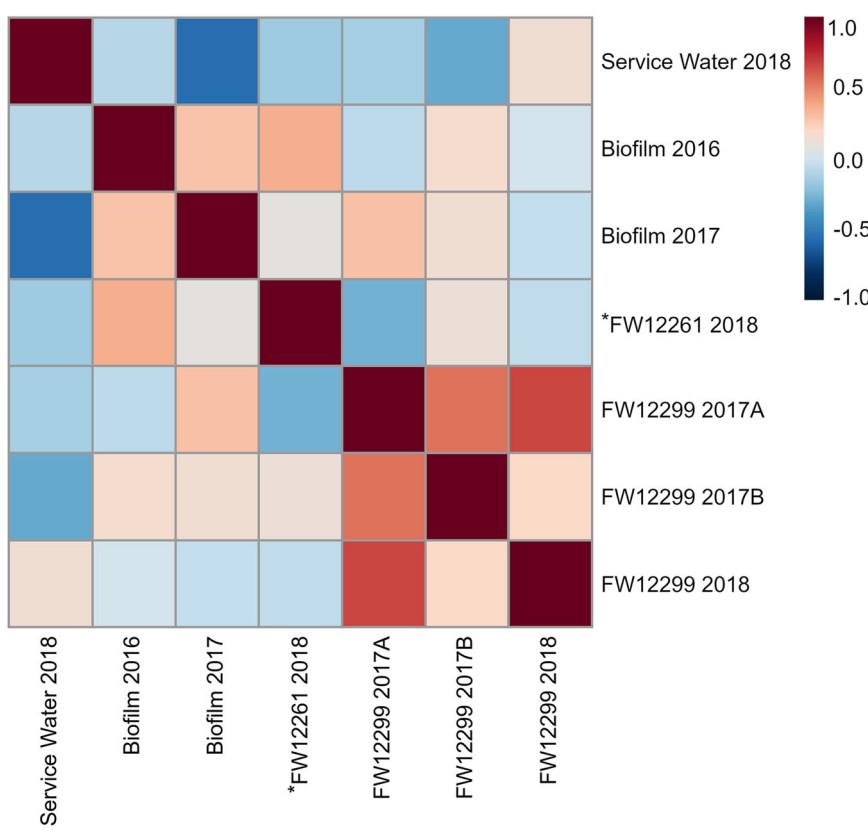

**Fig. 5 | Correlation heatmap for the microbial taxa ASVs of each biosampler, biofilm, and the service water generated using Spearman's rank correlation (ρ).** Where numbers are more positive (red), it indicates a strong, positive correlation. Numbers that are closer to zero indicate no correlation. More negative numbers (dark blue) are indicative of a weak negative correlation. The biosampler (*FW12261-2018) that became detached during sampling is designated with a star (*).

Supplementary Table 3). The 2018 biosampler FW12299-2018 had 17 total PLFA, with three making up 82% of the total abundance: 16:0, 18:0, and 18:1c$^9$ (Table 1, Supplementary Table 3). While the PLFA profiles were statistically different between the 2017 samples and the 2018 sample, the δ$^{13}$C values of the PLFA remained comparable for both years with an average δ$^{13}$C value of −29.5‰ (range: −30.3‰ to −27.8‰; n$_{PLFA}$ = 6) (Table 2). All FW12299 biosamplers were statistically different from the service water. Biosamplers FW12299-2017A and FW12299-2017B had PLFA profiles that were strongly correlated (ρ = 0.72, p < 0.01) (Fig. 3). The FW12299-2018 biosampler was weakly correlated to FW12299-2017B (ρ = 0.32, p < 0.01) (Fig. 3).

Importantly, the service water PLFA profile from 2018 was distinct from all biofilms and biosamplers. The service water sample had 18 PLFA, with three PLFA representing 88% of the total abundance: 16:0, 16:1$^x$, and 18:1t$^9$ (Table 1, Supplementary Table 3). The δ$^{13}$C value of service water PLFA was also distinct from values for the FW, with an average δ$^{13}$C$_{PLFA}$ of −34.6‰ (range: −39.0‰ to −30.9‰) (Table 2). The results from this environmental control effectively demonstrate that biofilms and fracture waters biosamplers are distinct from microbial communities related to service water contamination. The service water was distinct from all other samples (-0.29 ≤ ρ ≤ 0.013, p < 0.03 or 0.25 ≤ p ≤ 0.92) (Fig. 3).

## 16S rRNA Gene Analysis

A total of 482 amplicon sequence variants (ASVs) were identified across all samples with 30,642 to 62,838 reads obtained (Table S3). The most abundant ASVs were affiliated with *Candidatus* Frackibacter and *Sphingomonas echinoides* for the borehole and biofilms samples (Fig. 4, Table S3). The three ASVs associated with *Candidatus* Frackibacter (ASV 27, ASV #14, and ASV #165) found in the biosamplers were closely related and had 99.6% sequence similarity. These ASVs do not appear in the service water. In contrast, *Thiobacillus* sp. (32.05 ± 0.55%) and members of the family *Comamonadaceae* (20.5 ± 1.0%) were the two most abundant ASVs in both service water samples (Fig. 4). Consistent 16S rRNA gene profiles were

obtained for replicate analyses of the service water, which showed a more diverse profile than biofilm and fracture water samples, with the highest diversity of ASVs detected relative to all collected samples. The distinct nature of the service water results provides a critical control on contamination and allows the differences between the service water and the biosamplers that follow to be evaluated in the context of the indigenous microbial community characteristics. The taxa detected in each sample was treated as a profile or fingerprint of that community, allowing for analysis using Spearman's rank correlation to reveal pair-wise similarity of the samples based on Spearman's ρ and presented as a heatmap (Fig. 5). All community taxa profiles for the biosamplers and biofilms had no significant positive correlation in pairwise comparisons with the service water (Fig. 5).

In contrast to the service water, biofilm samples were dominated by ASVs affiliated with either *Sphingomonas echinoides* or halophilic species including *Candidatus* Frackibacter. The 2016 biofilm sample was dominated by *Candidatus* Frackibacter (49.5%, ASV# 27, and ASV# 14 and #165). in combination with *Halocalculus aciditolerans* (36.9%, ASV #134 and #262), *Haloferax sp.* (11.4%, ASV #206), and *Aliifodinibius sp.* (1.1%, ASV #332) (Fig. 4). In contrast, the biofilm sample from 2017 showed a much lower presence of *Candidatus* Frackibacter (1.3%, ASV #165), *Halocalculus aciditolerans* (7.5%, ASV #134), and *Haloferax sp.* (1.7%, ASV #206), and a high relative abundance of *Sphingomonas echinoides* (88.2%, ASV#27) which was not detected in the biofilm sample in 2016 (Fig. 4). The microbial taxa profiles from the biofilm samples had no positive correlation with each other, or with any of the biosamplers (-.064 ≤ ρ ≤ .36, 0.062 ≤ p ≤ 0.82) (Fig. 5).

Biosampler *FW12261-2018 was the only sample from this borehole with successful 16S rRNA gene amplification, despite repeated attempts to amplify FW12261-2017A and 2017B. This is likely due to the exceptionally high salinity in these samples. *FW12261-2018 had no significant correlation to any other samples (-.27 ≤ ρ ≤ .36, 0.063 p ≤ 0.81) (Fig. 5). This sample was dominated by *Sphingomonas echinoides* (84.4%, ASV #27) with smaller relative abundance of *Halocalculus aciditolerans* (5.1%m ASV

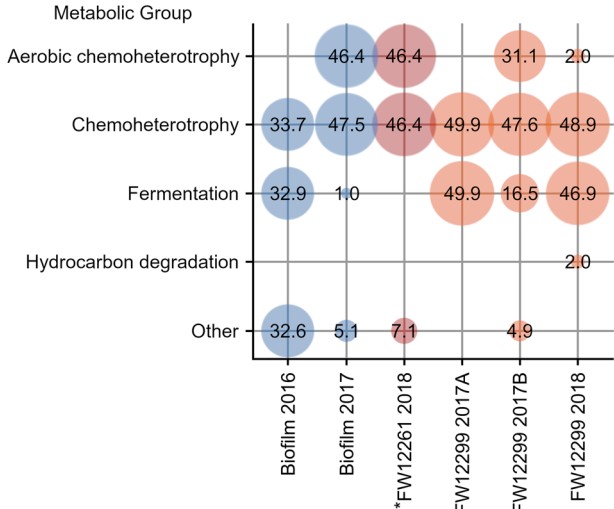

**Fig. 6 | Putative metabolic functions mapped using the FAPROTAX database to the microbial taxa presented in Fig. 4.** The biosampler that became detached during deployment (*FW12261 2018) is designated with a star (*).

#134), similar to the 2017 biofilm sample (Fig. 4). As mentioned previously, this biosampler had become disconnected from the borehole at an unknown point between 2017 and 2018.

While biosamplers from 2017 (FW12299-2017A and FW12299-2017B) had differences in ASV distributions, all samples from FW12299 show the same high abundance of *Candidatus* Frackibacter. FW12299-2017A was dominated by *Candidatus* Frackibacter (99.2%, ASV #14, #165, #593) with no detectable sequences from *Sphingomonas echinoides*. In contrast, FW12299-2017B biosampler was populated by a combination of *Sphingomonas echinoides* (59.1%, ASV #27) and *Candidatus* Frackibacter (31.3%, ASV #14, #165). In 2018, FW12299-2018 was again dominated by *Candidatus* Frackibacter (92.3%, ASV #165, #593), with only *Marinobacter* (3.9%, ASV #99) above 1% relative abundance (Fig. 4). The two 2017 biosamplers had moderately positive correlation with each other ($\rho = 0.55$, $p < 0.01$) (Fig. 5) while FW12299-2017A had significant correlation to FW12299-2018 ($\rho = 0.67$, $p < 0.01$) (Fig. 5). Both these latter two samples were significantly different from the biofilms and from *FW12261-2018 and, as noted above, completely distinct from the SW-associated ASVs.

## Metabolic Profiling with FAPROTAX

Putative metabolic capabilities were assigned with FAPROTAX, revealing that all fracture water and biofilm samples, in contrast to the service water, had the same predicted metabolic capabilities for taxa identified (Fig. 6). The community from the 2016 biofilm was assigned to fermentation (*Candidatus* Frackibacter, ASV #14 and #165) and general chemoheterotrophy (*Candidatus* Frackibacter, ASV #14 and #165). In the 2017 biofilm, *Candidatus* Frackibacter (ASV #165) was indicated to use chemoheterotrophy and *Sphingomonas echinoides* (ASV #27) used both aerobic chemoheterotrophy and general chemoheterotrophy. Most ASVs from *FW12261 2018 were predicted to be aerobic chemoheterotrophs (*Sphingomonas echinoides* ASV #27) and general chemoheterotrophs (*Chromohalobacter* ASV #186). The use of both fermentation and chemoheterotrophy by *Candidatus* Frackibacter was indicated for both FW12299 2017A (ASV # 14, 165, 593) and FW12299 2017B (ASV #14, 165). *S. echinoides*, detected only in FW12299 2017B, was implicated in both general and aerobic chemoheterotrophy (ASV #27). The FW12299 2018 community showed the use of fermentation and chemoheterotrophy by *Candidatus* Frackibacter (ASV #165, 594).

## Discussion

The goal of this study was to further characterize the microbial communities living within the ancient fracture waters of the KCO and determine the

carbon sources that may be supporting their growth. The combined analysis of geochemistry, PLFA, stable carbon isotopes, and nucleic acids, revealed evidence of two distinct microbial community patterns within the sample set. One community was dominated by halophilic bacteria from the family *Halobacteroidaceae*, *Candidatus* Frackibacter, that appeared to be carrying out anaerobic chemoheterotrophic and fermentative metabolisms (Fig. 6) and were observed primarily in FW12299. The second community consisted predominantly of *Sphingomonas echinoides* that were likely carrying out aerobic metabolisms using the dissolved organic carbon (DOC) within the system. This organism was observed in samples with greater exposure to the mine environment, primarily the 2017 biofilm sample and the biosampler that became disconnected from the borehole (FW12261 2018). Both microbial communities were statistically different from the mine service water that had their own distinct microbial community and geochemistry. Because the biosampler did not provide any nutritional substrate for microbial growth, any growth that did occur was interpreted to be in response to existing substrates present in the fracture fluids flowing through the sampler. Borehole 12299 has remained isolated from the mine environment by a packer for over a decade, while borehole 12261 had a packer installed only for the sampling period. Therefore, the microorganisms on the FW12299 biosamplers were likely more representative of indigenous microorganisms derived from the fracture water[43,45]. These lines of evidence support the interpretation that the fracture water communities, in particular FW 12299 from the borehole packered since 2012, are not derived from the external mine environment. Instead, the *Candidatus* Frackibacter dominated community is interpreted as being indigenous to the fracture fluids and host rock, as discussed in more detail below.

### Indigenous Origins of *Candidatus* Frackibacter

The presence of the putatively halophilic obligate anaerobe *Candidatus* Frackibacter in the most isolated samples from KCO suggests an indigenous origin for this member of the observed microbial communities. The absence of *Candidatus* Frackibacter in the low salinity mine service water supports this interpretation by excluding possible cross-contamination from the service water. Previous geochemistry studies of this system further corroborate this, having shown that there is no remnant component of modern surface recharge or drilling fluid contamination in these geologically isolated fracture waters based on $\delta^{18}O$ and $\delta^{2}H$ signatures, noble gas isotope signatures, and noble gas-derived mean residence times[23,31,43,48,52].

*Candidatus* Frackibacter has been identified in the terrestrial subsurface previously within black shale beds in the Appalachian Basin where its origin was attributed to the hydraulic fluids injected into the shale bed wells[62]. This inference was made despite not detecting any *Candidatus* Frackibacter in the injected hydraulic fluid. The authors assumed that it was below detection limit in the hydraulic fluids and bloomed only after injection. As a member of the phylum *Firmicutes*, the capacity of *Candidatus* Frackibacter to survive the inhospitable conditions in drilling fluids via endospores cannot be ruled out[63]. Daly et al.[62] suggested that *Candidatus* Frackibacter was unique to hydraulic fracking systems and sourced from the hydraulic fluids used to create fractures in the shale beds for hydrocarbon extraction. The findings of the present study suggest a re-evaluation of this hypothesis is warranted.

The service water used at Kidd Creek mine (i.e., recycled lake water) are very different from hydraulic fracking fluids, which are a chemical cocktail containing a variety of additives (e.g., glycine betaine, methanol, ethylene glycol), many of which are proprietary. In Kidd Creek *Candidatus* Frackibacter was not detected in the service water samples and was present at low abundances in samples that had greater exposure to the mine environment such as *FW12261-2018, as well as the biofilm samples, and thus these were eliminated as sources for this bacterium. The environmental controls employed by this current study provide a robust rationale to suggest the presence of *Candidatus* in this system cannot be attributed to contamination from either surface or mine service (drilling) waters. These lines of evidence suggest that the anaerobic halophile *Candidatus* Frackibacter is native to the Kidd Creek fracture waters system. It is possible that the earlier

interpretation from Daly et al.[62] was incorrect and *Candidatus* Frackibacter may also similarly be native to the Appalachian shales that were the focus of their study[62]. Despite differences in geologic age, there are common features in both study environments, including high salinity fluids hosted in fractured rock aquifers, and the presence of abundant carbon-rich layers in the fractured rocks. The discovery of *Candidatus* Frackibacter in Kidd Creek and its previous identification in unrelated surface settings[64], indicates that *Candidatus* Frackibacter is not unique to fractured shales or the hydraulic fluids thereof, and thus may be of broader biogeochemical interest in investigations of the deep subsurface biosphere than previously thought.

The ASVs for *Candidatus* Frackibacter in this study also return an NCBI BLASTN match to an uncultured bacterial isolate from a deep, relatively hot, saline aquifer in the North German Basin[65], with 99.4% nucleotide identity, though with only 87% query coverage. Those authors also reported the presence of a bacterium often regarded as an indicator of indigenous subsurface biome, Candidatus *Desulforudis audaxviator*[3,16,66], further establishing the indigenous nature of the biome under investigation there. The authors noted that the obligately anaerobic metabolism and high salinity tolerance of the bacteria they identified[60,67–69] make those bacteria well adapted for survival in the highly saline and anoxic fluids[65]. Taken together, these results suggest that *Candidatus* Frackibacter, and perhaps *Halobacteroidaceae* in general, may be features of naturally occurring subsurface microbial communities particularly when considering anoxic saline systems hosted within carbon-rich host rocks.

### *Sphingomonas echinoides* as an Indicator of Mine Exposure

The other distinct microbial community in this study is defined by *Sphingomonas echinoides*. The presence of large proportions of *S. echinoides* in some samples in this study was concurrent with increased exposure to the mine environment, comprising major proportions of \*FW12261-2018 (84.4%) which became detached from the packer during sampling, and the 2017 biofilm sample (88.2%). This organism was also present in FW12299-2017B (59.1%). Known *Sphingomonas* sp. are aerobic chemoheterotrophs commonly detected in the subsurface and have been previously used as a proxy for oxygen exposure in other oxic and microoxic subsurface environments[70,71]. Thus, the presence of *Sphingomonas echinoides* in a given sample in the context of the Kidd Creek fracture fluids appears to be indicative of exposure to the ambient mine environment. This interpretation is consistent with the presence of this bacterial population in biofilms sampled from the mine walls that were exposed to the mine environment, and in the sampler that became disconnected (\*FW12261-2018). In addition, it is important to note that in contrast to the 12299 borehole which has been isolated by continuous packering for more than ten years, the normal condition of 12261 prior to this sampling project was an open and freely flowing borehole. Previous work has shown such boreholes have fluids impacted by the effect of long-term interaction and contact with the mine air[43,45]. Previous analysis within the Kidd Creek system found that the microbial community of a freely flowing borehole in the Kidd Creek system similar to FW12261, FW12322, was also significantly different from that of FW12299[45]. The results of the current study, wherein the community from a freely flowing borehole, FW12261, is markedly different from one within a continuously packered system, FW12299, support the interpretation of an indigenous community in the packered FW12299, and a more impacted community reflected in the other samples.

Samples that contain both *Candidatus* Frackibacter and *S. echinoides* are an intermediate between the most isolated and mine-exposed communities. The 2017 biofilm, unlike the 2016 biofilm, had low relative abundance of *Candidatus* Frackibacter (1.3%) and other putatively anaerobic halophiles (Fig. 3). This may reflect a gradient of geochemical conditions occurring due to partial exposure to the mine environment. Such a gradient would be expected for a biofilm community since the variable flow of fracture water could have been insufficient to fully inundate the biofilm. This is likely given the high relative abundance of *S. echinoides* (88.2%) (Fig. 3). Gradients of available geochemistry including oxygen are a hallmark of biofilm structure, and so a heterogeneous community is more likely.

**Table 3 | Δδ13C[CS]-PLFA (‰) values for the two PLFA signatures from the detected Kidd Creek microbial communities. The carbon sources (CS) are indicated, with the δ13C (‰) values determined previously in the indicated references**

| | FW12299 | | FW12261 | |
|---|---|---|---|---|
| | δ$^{13}$C (‰) | Δδ$^{13}$C$_{PLFA - [CS]}$ (‰) | δ$^{13}$C (‰) | Δδ$^{13}$C$_{PLFA - [CS]}$ (‰) |
| PLFA | −14 | – | −28 | – |
| DIC[24] | −0.5 | −13.5 | −8.8 | −19.2 |
| DOC[24] | −8.4 | −5.6 | −5.1 | −22.9 |
| Formate[24] | −17.7 | 3.7 | −9.7 | −18.3 |
| Acetate[24] | −6.9 | −7.1 | −5.3 | −22.7 |
| CH$_4$$^{avg}$[51] | −34 | 20 | −38 | 10 |
| SOC$^{max}$[121] | −19 | 5 | −19 | −9 |
| SOC$^{avg}$[121] | −26 | 12 | −26 | −2 |
| SOC$^{min}$[121] | −33 | 19 | −33 | 5 |

*DIC* dissolved inorganic carbon, *DOC* dissolved organic carbon, *SOC* sedimentary organic carbon.

It may be that the biofilm samples collected in the different years captured different components of the biofilm.

Sample FW12299-2017B had high proportions of both *S. echinoides* (59.1%) and *Candidatus* Frackibacter (31.3%) (Fig. 3). This sample was deployed in tandem with FW12299-2017A with the gas and water flow split between the two biosampler inflows and it was expected that these two biosamplers would have the same community composition. Indeed, the PLFA distribution and δ$^{13}$C$_{PLFA}$ values from both FW12299-2017A and FW12299-2017B are highly similar, however the 16S rRNA analysis shows two starkly different community compositions. The mixed community detected in FW12299-2017B is likely representative of a fracture water derived community that has been exposed to the external mine environment. This is likely due to both the variable flow of water and gas from the borehole and its subsequent splitting between the two tandem biosamplers.

### Regional Sedimentary Carbon Supports the Most Isolated Microbial Communities Sampled in Kidd Creek Mine

The detection of *Candidatus* Frackibacter and its indigenous status within the Kidd Creek fracture waters prompts questions regarding the metabolic strategies it may be employing. Daly et al.[62] suggest that *Candidatus* Frackibacter is consuming the glycine betaine present in the hydraulic fluids injected into the fracking wells; however, this additive is not present in the Kidd Creek waters. To investigate the putative metabolic capabilities of this bacterium, the stable carbon isotopic signature of the PLFA (δ$^{13}$C$_{PLFA}$) was determined and compared to the δ$^{13}$C of potential carbon sources. Subsequently, FAPROTAX analysis of the *Candidatus* Frackibacter ASVs provides insight into and support for the δ$^{13}$C results.

The δ$^{13}$C$_{PLFA}$ had two groups of samples with distinct signatures of circa −28‰ (FW12299-2017A, FW12299-2017B, FW12299-2018, and \*FW12261-2018) and circa −14‰ (FW12261-2017A and B; 2016 biofilm; 2017 biofilm) (Table 3, Supplementary Table 3). This suggests that there are different metabolic pathways being employed by the communities between these two sets of samples. A comparison of the δ$^{13}$C$_{PLFA}$ with that of potential carbon sources (δ$^{13}$C$_{CS}$), Δδ$^{13}$C$_{PLFA-CS}$, can be used to assess metabolic pathways and carbon source utilization based on published ranges [e.g.[72,73]]. The organic carbon sources present within the Kidd Creek system include abiotically produced methane (δ$^{13}$C$_{CH4}$ = −31.9 to −42.2‰)[54,56,61], dissolved organic carbon (DOC δ$^{13}$C$_{DIC}$ = −5.1 to −8.9‰), dissolved inorganic carbon (δ$^{13}$C$_{DIC}$ = −0.5 to −8.8‰)[24] and carbon-rich host rocks (δ$^{13}$C = −18 to −33‰)[61]. The DOC concentrations have been previously reported for 12261 and 12299 as 2400 μM and 5000 μM, respectively, of which up to 90% was acetate and formate[24]. The Δδ$^{13}$C$_{PLFA-CH_4}$ indicate that the PLFA are $^{13}$C-enriched relative to the

methane which excludes methane as a carbon source despite its abundance because methanotrophy produces highly depleted PLFA relative to methane (Table 3)[54,56,73–78]. Warr et al.[54] showed evidence that methanogenesis and methanotrophy may be occurring within the Kidd Creek system but at such low rates and conversion that evidence was only visible via very sensitive clumped methane isotope analysis. The $\Delta\delta^{13}C_{PLFA-CH_4}$ values determined in this study also suggest only minimal biological methane cycling as values are all positive (Table 3), where expected fractionation from biological cycling of methane is typically very large and negative[74–78].

The $\Delta\delta^{13}C_{PLFA-CS}$ value for heterotrophy is generally considered to be small, but still variable, between −2‰ and −8‰ for aerobic communities, depending on substrate and metabolic pathway[79–86], but as large as −21‰ in anaerobic communities[86–88]. The DOC pool within both FW12261 and FW12299 were previously demonstrated to be dominated by acetate and formate, together accounting for between 50% to 90% of the total DOC[24]. The exceptionally high mM concentrations of acetate and formate, the unusual 2:1 acetate/formate ratio, and their extraordinarily enriched $\delta^{13}C$ signatures were hypothesized to be related to production of these organic acids by radiolysis, previously shown to account for both $H_2$[58] and dissolved $SO_4$[23] in this setting. The $\delta^{13}C_{PLFA-acetate}$ value of the community with the $\delta^{13}C_{PLFA}$ of −14‰ is consistent with the isotopic fractionation expected for aerobic heterotrophic utilization of acetate by these communities that have experienced greater exposure to the mine environment. In contrast, utilization of acetate through anaerobic heterotrophy by the microbial communities with the $\delta^{13}C_{PLFA}$ of −28‰ is not indicated by the $\Delta\delta^{13}C_{PLFA-Acetate}$ value, −23.7‰, as it is not within the expected range of −5‰ to −21‰[86–88]. Additionally, Sherwood Lollar et al.[24] demonstrated that the abiotically produced acetate is not likely being consumed within the Kidd Creek fracture waters, in part based on an absence of any variation in acetate concentrations and associated $\delta^{13}C$ signatures over a decade of monitoring. These observations are consistent with the fact that the dominant organism within the group of samples with $\delta^{13}C_{PLFA}$ = −28‰, Candidatus Frackibacter, is considered to be an acetogen[62] and so catabolism of acetate is unlikely.

This lack of utilization of acetate and methane suggests that Candidatus Frackibacter is utilizing the meta-sedimentary organic carbon (SOC) in the regional setting where Kidd Creek is located. The $\delta^{13}C$ of the SOC in the Superior Province of the Canadian Shield, including surface exposures, has a wide range (−17.5‰ to −43.8‰) due to variability across the entire area[89–91]. The total organic carbon (TOC) in Kidd Creek itself has a more limited range of −18‰ to −33‰, while the carbonaceous argillites at Kidd Creek have a narrower range of −19‰ to −26‰[89]. While the $\delta^{13}C$ of the specific units that are intersected by 12299 and 12261 is not known, the literature values can be used to generate a general range expected for the $\delta^{13}C$ of the organic carbon in Kidd Creek that can be summarized as: $\delta^{13}C_{SOC}^{Max}$ = −18‰; $\delta^{13}C_{SOC}^{Avg}$ = −26‰; $\delta^{13}C_{SOC}^{Min}$ = −33‰. Using these values, the range of $\Delta\delta^{13}C_{SOC-PLFA}$ values for the microbial community within the samples with $\delta^{13}C_{PLFA}$ = −28‰ is calculated to be +4‰ to −11‰ (Table 3) which, except in the case of the most enriched portion of the SOC pool ($\delta^{13}C_{SOC}^{Max}$), falls within the range for anaerobic heterotrophy[79–81]. Multiple studies have demonstrated that even refractory organic carbon layers can support indigenous microbial communities at the Earth's surface[92–94], in the coal or shale beds of deep sedimentary basins [e.g.[95].], and in deep fracture waters in ancient Precambrian Shield settings[19,24,96–107]. Particularly in the subsurface, processes such as metasomatism, thermal decomposition, and radiolysis of the carbon-rich layers in the rock can release organic carbon from sedimentary deposits that provide sources of carbon (including methane and volatile fatty acids) to the dissolved organic matter (DOM) in associated fracture fluids[19,20,24,89,108,109], just as black shales in younger sedimentary basins are known to be a source of carbon to surrounding groundwater microbial communities due to diagenesis and fermentation[92,93,106,107,110]. Several past studies have investigated what compounds are released to the dissolved organic matter (DOM) from the refractory carbon layers common in the host rocks in this part of the Canadian Shield. Ventura et al.[111] demonstrated from solvent extractions in

2.7 billion year old rocks near Timmins that a range of biphytanes and hopanes as well as unresolved complex mixtures are common in these ancient hydrothermal settings on the Canadian Shield. In similar ancient settings in the Witwatersrand Basin of South Africa, two studies used water-soluble extracts and direct analysis of DOM to identify not only microbial metabolites contributing to the DOM but evidence of older fractions likely originating from the carbon-rich refractory layers common also in the gold mines there[20,109]. Nisson et al.[19,33] suggested that radiolysis might be reworking refractory carbon from the host rock layers and thereby producing a DOM pool that would be more bioavailable to the microbial communities documented throughout the Witwatersrand basin gold mines[23]. Given the large role of radiolysis demonstrated for the Kidd Creek fluids in previous studies[23,58], such processes might also be a mechanism for delivering carbon-bearing substrates form the rocks to the DOM pool. Alternatively, Candidatus Frackibacter could be metabolizing organic carbon from the geologic system via fermentation or heterotrophy of the organic compounds observed by Ventura et al.[111], and in fact, the limited contribution of larger organic compounds to the DOM[24] could indicate their consumption.

Regardless of the source, use of such a DOM pool is consistent with the known metabolic capabilities of Candidatus Frackibacter, the functional predictions from 16S rRNA genes in this study which suggest chemoheterotrophy and fermentation (Fig. 6), metabolic capabilities suggested by comparison to previous studies of the KCO fracture waters. Notably, Wilpiszeski et al.[45] identified organisms related to Candidatus Frackibacter in their analysis of FW12299 including Fuchsiella ferrireducens. The fact that they did not report detection of Candidatus Frackibacter may be due to differences in available genetic databases for comparison. The representative sequences for the Candidatus Frackibacter ASVs detected in this study were subject to a BLAST search, returning a > 95% similarity to the Fuchsiella ferrireducens Z-7101T (NR_136778.1; NR_136777.1; NR_136776.1) and Z-7102 (KM215211.1)[68,69]. F. ferrireducens is a haloalkaliphilic homoacetogen that is capable of iron reduction and may also be able to respire acetate under anoxic conditions, using sulfate as an electron acceptor as done by sulfate-reducing bacteria[68,112–114]. Other microorganisms found at >95% similarity in this BLAST search include Fuchsiella alkaliacetigena (Accession NR_118038.1)[69] and the halophilic and selenate-respiring anaerobe Selenohalobacter shriftii (NR_028804.1)[115]. Candidatus Frackibacter, F. ferrireducens, and F. alkaliacetigena are all members of the family Halobacteroidaceae, indicating that similar metabolic lifestyles are possible. It may be possible that, in addition to fermentation, Candidatus Frackibacter may also possess the ability to carry out sulfate or iron reduction, consistent with other members of this family. However, the variability of this potential metabolic pathway presently remains unclear as the metabolic capabilities of Candidatus Frackibacter have not been fully characterized to date. Such utilization of the SOC and/or acetate in combination with the sulphate present in the Kidd Creek system present is consistent with evidence of a long-standing sulfur cycle in the Kidd Creek system[23] and with previous most probable number analysis of sulfate-reducing and alkane-oxidizing bacteria[43].

The findings of this study suggest that rather than an artifact of drilling activities, Candidatus Frackibacter may have a range that is more geographically widespread than previously thought, ranging from sedimentary basins to crystalline shield rocks. The possible use of refractory organic carbon by Candidatus Frackibacter also implies that this organism may be able to persist within isolated subsurface systems over long, geologically relevant time periods. By demonstrating that microorganisms can and do survive deep burial and subsequent isolation from the surface on geologically relevant timescales through the utilization of rock-derived carbon, a better understanding of habitability can be achieved. This holds implications for the habitability of all isolated deep subsurface fluids, including those that are found extraterrestrially, such as within the subsurface of Mars, or the subsurface oceans of Europa and Enceladus.

## Conclusions

Samples collected from a highly isolated subsurface microbial community from the KCO were dominated by a single member of the bacterial family

*Halobacteroidaceae*: *Candidatus* Frackibacter. This bacterium may be adapted to living in hypersaline, anoxic, and nutrient-limited waters, potentially on geologically relevant timescales, and may be indigenous to the subsurface biosphere. In contrast to current expectations, these results demonstrate that *Candidatus* Frackibacter is not unique to hydraulic fracking systems as previously proposed[62]. The Kidd Creek fracture waters investigated here may represent an ancient analogue to groundwaters in shale bed systems, as the Kidd Creek habitat shares four common and important features with modern fracking systems: abundant carbon rich rock, fractured rock, anoxic conditions, and saline fluids. The presence of aerobic *Sphingomonas* taxa in some samples was interpreted as an indicator of impact or introduction of organisms from the mine environment. Based on the $\delta^{13}C$ patterns and inferred metabolisms using FAPROTAX the *Candidatus* Frackibacter population living in the most isolated samples likely utilizes mixed anaerobic chemoheterotrophy and/or fermentation wherein the active metabolism is determined by availability and abundance of local carbon-sources. If bacteria such as *Candidatus* Frackibacter are found in many of these deep subsurface communities then, like *Desulforudis audaxviator*, further study of these taxa may reveal important identifiers of deep subsurface life. Characterizing these identifiers is crucial in understanding the colonization of the deep subsurface, how long life survives in the deep biosphere, and what this may mean for understanding the evolution of the deep Earth. These results additionally imply that similar deep subsurface environments may offer a refuge for life to survive over long periods, including in the type of saline waters thought to be present within the subsurface of Mars or the subsurface oceans of Enceladus and Europa.

## Methods
### Site description
The sampling sites are located at 7850 ft (2.39 km) below the surface in the KCO, within a working copper-silver-zinc mine, in Timmins, Ontario, Canada. Samples were collected from two uncased boreholes with identifiers 12299 and 12261, positioned approximately 1 m apart. Borehole 12299 has a length of 603 m and 12261 has a length of 1422 m. The geochemical composition and age of isolation of the water flowing from these boreholes is described in detail elsewhere[23,43,48,52], and summarized briefly here. Quantification of noble gas components produced via radiogenic in-growth within a closed system ($^4He^*$, $^{21}Ne^*$, $^{40}Ar^*$, $^{136}Xe^*$) revealed that the deep bulk fracture fluids of Kidd Creek have a mean residence time of approximately 1.5 billion years[48], with youngest fluid contributions of 150–650 Ma in age[52]. This finding was reinforced by the presence of an early atmospheric xenon signature ($^{124-128}Xe$) coupled with water isotopic data indicative of hydrogeologic isolation over Ma-Ga timescales[31,48,52]. Previous geochemical analysis indicated that the fluids are highly saline brines, with salinities on the order of 3.6 M, contain methane, hydrogen, and sulfate produced by water-rock reactions[23,31,53,54,56] and can support microbial growth[43,45,54].

### Biosampler preparation and installation
The biosamplers consisted of cylindrical housings made of 316 grade (low carbon) stainless steel that were packed with 21.3 g ($\pm 1.7$ g) of carbon-free and aluminum-coated glass wool in the main body, with 2.25 g of uncoated glass wool placed in the housing endcaps. Complete housing units were fitted with 3/8" hose barbs and the entire unit was wrapped in aluminum foil and combusted in a Thermolyne 30400 furnace at 450 °C for 8 h for sterilization as previously described[116]. Due to the long installation time in the boreholes (~ 6–8 months), biomass collected from the biosamplers represents microorganisms filtered from the water, as well as any in situ growth of microorganisms on the biosampler matrix (Fig. 1). Because the biosampler matrix only provides an attachment surface, any growth that occurred within the matrix is attributable to microbial communities and the substrates naturally present in the fracture fluids flowing through the biosampler.

The biosamplers were connected to Margo-style grout plugs using solvent-rinsed 3/8" Tygon tubing and metal hose-clamps. Where needed, 1/4" tubing and a reducing union were used to facilitate attachment of the housings. Downstream of the units, 3/8" tubing was used to attach flow rate monitors at the outflow. During sampling in 2017, tandem biosamplers were deployed for each borehole. This involved splitting the outflow from the installed manifolds and directing it to two biomass collection units for each borehole. These biosampler units were secured in place horizontally, one on top of the other (and designated 2017A and 2017B).

In total, six biosamplers from two boreholes, two samples of biofilm growing adjacent to borehole 12261, and one service water sample were collected between July 12, 2016, and January 29, 2018 (Fig. 1). In addition, gas and water flow rates, temperature, and pH were measured (Supplementary Table 1). Service water was monitored for geochemistry at all sampling points (Supplementary Table 2). Only the January 29, 2018, service water sample was used for PLFA and DNA analysis. Prior to retrieval in 2018, the biosampler attached to borehole 12261 detached from the borehole for an unknown amount of time. This biosampler was given a star-designation to indicate this, *FW12261-2018, and was processed along with the others and provides information on the impact of the surrounding mine environment on these samples. Biofilms were also sampled aseptically from the rock face adjacent to 12261 by using an inverted Whirlpak and solvent rinsed stainless-steel spatula to manually detach a portion of the biofilm from the wall. The Whirlpak was then turned right-side-out around the biofilm. The mine service water is piped into the mine from a surface lake and had previously been geochemically and isotopically characterized[43]. For this study 17.3 L was collected from a service water valve at the sampling site into a sterile carboy to characterise the microbiology of this potential source of contamination. The entire volume was vacuum filtered at the surface through 0.22-μm filters in 2018. Due to high abundance of particulates in the service water, multiple filters were used. Two filters were divided such that one half was reserved for DNA analysis, and the other extracted for PLFA with the other pooled filters.

### DNA Extraction, 16S rRNA Gene Amplification, and Sequencing
Of the total biosampler substrate, 10% by mass was collected from multiple locations across the biosampler material to form a composite sample for DNA analysis to account for spatial variability in a low biomass sample. Freeze-dried biofilm samples were homogenized using a sterilized glass rod. Of each biofilm, 200 mg was used for DNA analysis. Prior to DNA extraction, the freeze-dried biofilm and biosampler samples were washed with three times with 50 mL of PCR-grade water to reduce salt content and sonicated to free the cells from the accompanying matrix. Subsequently, the waters collected were filtered through a 0.22-μm filter to reduce salt content and isolate biological material. The filter paper from the service water 2018 sample was prepared in duplicate to ensure accurate representation of this possible source of contaminant microorganisms. Filters were stored at −20 °C until DNA extraction. The DNA was extracted from randomized subsamples using the DNeasy PowerSoil Kit (Qiagen). Filter samples were cut into small pieces using a DNA-free single-use scalpel and placed into the PowerBead tube. After addition of the lysis solution C1, the PowerBead tubes were incubated at 70 °C for 10 min in a rotating incubator, followed by 45 s of bead beating at 5.5 m/s using the FastPrep instrument (MP Biomedicals). The remainder of the extraction was carried out following manufacturer protocol. Extracted DNA was eluted into 60 μL of solution C6 (10 mM Tris buffer). DNA concentrations were determined using the Qubit dsDNA High Sensitivity Assay kit (Invitrogen, CA, USA) with fluorescence measured on a Qubit 4.0 fluorometer (Life Technologies, CA, USA). Concentrations were below detection limit for all samples except the service water (21 ng/μl), biofilm 2017 (4.6 ng/μl), biofilm 2016 (2.5 ng/μl), and *FW12261-2018 (0.06 ng/μl). While this may have resulted in the exclusion of minor components of the microbial communities, the most abundant taxa would still be captured. Controls included were field blanks, a filter blank, and a DNA extraction kit control. Additionally, no-template controls (NTCs) and positive controls were included during the PCR step. DNA concentrations of controls were below the detection limit of the Qubit dsDNA High Sensitivity Assay Kit (Invitrogen).

The 16S ribosomal RNA (16S rRNA) genes were amplified from randomized samples using universal prokaryotic primer 515F-Y/926 R[117,118].

Each primer contained a six-base index for multiplexing, Illumina flow cell binding, and sequencing sites[119]. The PCR mix (25 μL) contained 1X ThermoPol Buffer, 0.2 μM forward and reverse primer, 200 μM dNTPs, 15 μg bovine serum albumin, 0.625 U Hot Start Taq DNA polymerase (New England Biolabs), and 1 μL of template. Each PCR was prepared in triplicate and amplified as follows: 95 °C for 3 min, 40 cycles of 95 °C for 30 s, 50 °C for 30 s, 68 °C for 1 min, and a final extension of 68 °C for 7 min. Triplicate PCR products were pooled, quantified on a 1% agarose gel, and equal quantities of each PCR product were pooled. Biosampler samples FW12261-2017A and FW12261-2017B did not yield a PCR product but 5 µl was added to the sequencing pool nonetheless. The NTCs, filter blanks, field blanks, and kit controls were included in the sequencing pool using 5 µL volumes, even though no PCR product was visible in the agarose gel. The pooled 16S rRNA PCR products were excised from an agarose gel and purified using Wizard SV Gel and PCR Clean-Up System (Promega, WI, USA).

A total of 10 samples and 17 controls were sequenced as an 8 pM library containing 15% PhiX on a MiSeq instrument (Illumina, San Diego, USA) using a 2 ×250 cycle MiSeq Reagent Kit v2 (Illumina Canada). Reads were demultiplexed using MiSeq Reporter software version 2.5.0.5 and analyzed using Quantitative Insights Into Microbial Ecology 2 (QIIME2; version 2021.2)[120] managed by Automation, eXtension, and Integration Of Microbial Ecology (AXIOME3)[121]. DADA2 (version 2020.6)[122] was used to remove primer sequences, truncate forward (250 bases) and reverse (243 bases) reads, remove low quality reads and chimeras, as well as denoise and dereplicate the dataset. The AXIOME3 pipeline assigned taxonomy to amplicon sequence variants (ASVs) using a naïve Bayes classifier (classify-sklearn) trained on the SILVA 138 release[123]. Samples FW12261-2017A and FW12261-2017B did not yield sufficient reads and these samples were not included in downstream analyses.

Negative controls (field blank, filter blank, DNA extraction kit, NTC) were used to identify contaminant sequences using the prevalence method with a score statistic threshold value of 0.5 in Decontam (version 1.10.0)[124]. A PCR positive control, which contained the 16S rRNA genes of *Thermus thermophilus* and *Aliivibrio fischeri*, was included as well. A total of 8 ASVs were identified as contaminants by Decontam. All contaminant ASVs identified by Decontam were absent from samples (Data not shown).

FAPROTAX (version 1.2.4)[125] was used to assign putative metabolic functions to ASVs. Taxa without a known function were grouped into a category called "other". Functional profiles were created for ASVs at or above 1% relative abundance to show metabolic functions of most abundant ASVs within each sample. Sequences are available in the European Nucleotide Archive under accession number PRJEB49237[126].

### Phospholipid fatty acid extraction, quantification, and δ¹³C determination

Of the total mass of biosampler substrate, the remaining 90% of each sample was analysed for PLFA analysis. All biosamplers, biofilm, and 0.22-µm service water filter samples were freeze-dried prior to extraction for PLFA at the Environmental Organic Geochemistry Laboratory (EOGL) based at McMaster University, Canada. A modified protocol[127,128] with conversion to fatty acid methyl esters (FAMEs) by methanolysis[129] was used. The recovered FAMEs were separated using gas chromatography-mass spectrometry (GC-MS) with an Agilent GC-MS instrument (Agilent Technologies, Santa Clara, California, USA). FAMEs were separated on a DB5-ms+DG capillary column (30 m, 0.25 mm, 0.25 µm) using a GC temperature program of 50 °C (1 min), 10 °C/min to 160 °C, 1.5 °C/min to 180 °C (10 min), 1.5 °C/min to 250 °C, 20 °C/min to 320 °C. The PLFA identities were determined using retention time and mass spectra of standards (Bacterial Acid Methyl Esters Mix, Matreya, Pleasant Gap, Pennsylvania, USA). The PLFAs were named by the number of carbons:number of double bonds. Where known, cis- and trans- bond positions are indicated by "c" and "t", respectively. Iso- and anteiso- are denoted by "i" or "a", respectively. The PLFAs with unknown branch location were denoted with "br", and those containing cyclic groups are denoted with "cy". Double bond position was indicated by a superscripted number, and where exact double bond position is unknown a

superscripted letter (x or x, y) was used. All PLFA data are available through the EarthChem Library[130].

Determination of δ¹³C values for extracted PLFA was determined at the EOGL (McMaster University) using an Isotope Ratio-Mass Spectrometer (IR-MS) with a GC combustion configuration (Agilent 6890 N GC; GC Combustion III Interface; Delta Plus XP IRMS). The δ¹³C of the methanol used for methanolysis was used for the methylation measure and had a value of −37.4‰. The FAMEs were separated using a DB5-MS capillary column (30 m × 0.32 mm × 0.25 µm) with a GC temperature program of 80 °C (1 min), 4 °C/min to 280 °C, and 10 °C/min to 320 °C (15 min). The splitless injection port was held at 310 °C, the oxidation oven was 980 °C, and the reduction oven was 650 °C. Samples were run as a single analysis. The measured isotopic composition was expressed as δ¹³C (with total uncertainty incorporating both accuracy and reproducibility of ±0.5‰) and is defined in Eq. (1) versus V-PeeDee Belemnite. All δ¹³C data are available through the EarthChem Library.

$$\delta^{13}C(‰) = [(^{13}C/^{12}C)_{sample}/(^{13}C/^{12}C)_{standard} - 1] \times 10^3 \qquad (1)$$

### Statistical analysis

Geochemical data for all fracture and service water samples were analyzed using a one-way parametric ANOVA. All geochemical data is available through the EarthChem Library. The sum normalized PLFA and 16S rRNA data for each sample were analyzed by Spearman's rank correlation (ρ) with a $p$-value threshold of $p \leq 0.05$ to evaluate statistical differences that may exist. Spearman's ρ is a nonparametric measurement of the statistical dependence between the ranking of sample pairs and describes how well this relationship fits a monotonic function. All statistical analysis was performed using the MetaboAnalyst online portal[131–141].

### Reporting summary

Further information on research design is available in the Nature Portfolio Reporting Summary linked to this article.

### Data availability

The data for this project has been made available in the following repositories. Phospholipid fatty acid, stable carbon isotope, and geochemical data has been made available in the EarthChem Library cited as Sherwood Lollar, B., Slater, G. F., Engel, K., Warr, O., Lollar, G. S., Brady, A., Neufeld, J. D., 2024. Compiled geochemistry and PLFA data on fracture waters, biofilms and service waters from Kidd Creek Observatory, Version 1.0. Interdisciplinary Earth Data Alliance (IEDA). https://doi.org/10.60520/IEDA/113522. Accessed 2024-11-19. 16S rRNA sequence data is available through the European Nucleotide Archive under accession numbers PRJEB49237.

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

## Acknowledgements
Samples for this study were obtained from the traditional territory of the Anishinabewaki and Cree. Site access and permission for sample collection was provided by the landowners. Thanks are due to colleagues and supporters at the mine whose efforts and support for the sampling program were invaluable. BSL is CIFAR Fellow and Director and acknowledges the CIFAR Earth 4D Subsurface Science and Exploration Program, with additional funds from NSERC Herzberg Prize funds (506250). This research was supported by a Natural Sciences and Engineering Research Council of Canada (NSERC) Special Research Opportunities Grant to G.S. and B.S.L. (486080), an NSERC Alliance grant to J.D.N. and G.F.S. (567161-21), and NSERC Discovery grants to B.S.L. (453949), and G.F.S. (05456). Additional research funds and open access publication costs were from NSERC Herzberg funds to B.S.L. (506250).

## Author contributions
This work was directly supervised by G.F.S. S.E.F., G.F.S., O.W., and A.B. contributed to field work, sample collection, and sample transport. S.E.F. conducted all sample processing and instrumentation performed at the EOGL (McMaster University) with access and supervision provided by G.F.S. Analysis performed at the University of Waterloo was completed by K.E. and S.E.F., with access and supervision provided by J.D.N. B.S.L formulated original study design, along with G.F.S., and provided access to the field site through partnership with the Kidd Creek Mine. All coauthors (S.E.F., G.F.S., K.E., O.W., G.S.L., A.B., J.D.N. and B.S.L.) contributed to the interpretation of the results. SEF took the lead on manuscript writing and all authors (S.E.F., G.F.S., K.E., O.W., G.S.L., A.B., J.D.N. and B.S.L.) provided critical feedback and contributions to shaping the research, analysis and manuscript.

## Competing interests
The authors declare no competing interests.
