## [Transparent Peer Review file · Communications Earth & Environment]

Deep terrestrial indigenous microbial community dominated by *Candidatus Frackibacter*

Corresponding Author: Dr Greg Slater

Version 0:

Decision Letter:

Dear Dr Slater,

Your manuscript titled "Deep terrestrial indigenous microbial community dominated by *Candidatus Frackibacter*" has now been seen by 3 reviewers, and we include their comments at the end of this message. They find your work of interest, but some important points are raised. We are interested in the possibility of publishing your study in *Communications Earth & Environment*, but would like to consider your responses to these concerns and assess a revised manuscript before we make a final decision on publication.

We therefore invite you to revise and resubmit your manuscript, along with a point-by-point response that takes into account the points raised. Please highlight all changes in the manuscript text file.

In particular, here are a few thresholds that we would like you to meet with your revision:

- 1] One reviewer suggested an approach of deeper sequencing to more securely conclude metabolic characteristics of *C Frackibacter*. If such data are available, supporting your claims with those data is requested. If deeper sequencing data (or other additional evidence) is not available, we require you to instead please fully justify the conclusions concerning the metabolism and ecology of *C Frackibacter*, or soften the significance of the findings accordingly.
- 2] The presentation of the lipid data should include a description of the predominance of certain fatty acids in each type of sample and consider the suggested revisions to the display of the lipid data in the tables and figures.
- 3] In addition, all three reviewers had comments about the low concentration of some of the retrieved DNA, so please be fully transparent about what was retrieved and the possible impact the low concentrations might have on the interpretation of the results.

Please use the following link to submit your revised manuscript, point-by-point response to the referees' comments (which should be in a separate document to any cover letter), a tracked-changes version of the manuscript (as a PDF file) and the completed checklist:

Link Redacted

We hope to receive your revised paper within six weeks; please let us know if you aren't able to submit it within this time so that we can discuss how best to proceed. If we don't hear from you, and the revision process takes significantly longer, we may close your file. In this event, we will still be happy to reconsider your paper at a later date, as long as nothing similar has been accepted for publication at *Communications Earth & Environment* or published elsewhere in the meantime.

Please do not hesitate to contact us if you have any questions or would like to discuss these revisions further. We look forward to seeing the revised manuscript and thank you for the opportunity to review your work.

Best regards,

D'Arcy Meyer-Dombard, PhD
Editorial Board Member
Communications Earth & Environment
orcid.org/0000-0001-9862-4839

Joe Aslin
Deputy Editor
Communications Earth & Environment

EDITORIAL POLICIES AND FORMATTING

Editorial Policy: [Policy requirements](https://www.nature.com/documents/nr-editorial-policy-checklist.pdf) (Download the link to your computer as a PDF.)

- Behavioural and social science
- Ecological, evolutionary & environmental sciences
- Life sciences

<https://www.nature.com/documents/nr-reporting-summary.zip>

Furthermore, please align your manuscript with our format requirements, which are summarized on the following checklist: [Communications Earth & Environment formatting checklist](https://www.nature.com/documents/commsj-phys-style-formatting-checklist-article.pdf)

and also in our style and formatting guide [Communications Earth & Environment formatting guide](https://www.nature.com/documents/commsj-phys-style-formatting-guide-accept.pdf) .

*** DATA: Communications Earth & Environment endorses the principles of the Enabling FAIR data project (<http://www.copdess.org/enabling-fair-data-project/>). We ask authors to make the data that support their conclusions available in permanent, publically accessible data repositories. (Please contact the editor if you are unable to make your data available).

All Communications Earth & Environment manuscripts must include a section titled "Data Availability" at the end of the Methods section or main text (if no Methods). More information on this policy, is available at <http://www.nature.com/authors/policies/data/data-availability-statements-data-citations.pdf>.

If a community resource is unavailable, data can be submitted to generalist repositories such as [figshare](https://figshare.com/) or [Dryad Digital Repository](http://datadryad.org/). Please provide a unique identifier for the data (for example a DOI or a permanent URL) in the data availability statement, if possible. If the repository does not provide identifiers, we encourage authors to supply the search terms that will return the data. For data that have been obtained from publically available sources, please provide a URL and the specific data product name in the data availability statement. Data with a DOI should be further cited in the methods reference section.

REVIEWER COMMENTS:

Reviewer #1 (Remarks to the Author):

This research is novel and extends the information on isolated highly saline, deep subsurface bacterial communities in the Kidd Creek Observatory. Comparison of the more anaerobic fracture waters with biofilm and service water samples helps distinguish these isolated bacterial communities and relate their DNA and PLFA compositions to carbon sources and metabolic processes. The paper is well written but there is room for improvement and clarification in some areas prior to publication.

Please note that “indigenous” is misspelled on cover page.

P2 line 24, We suggest switching microbial to bacterial. And why:

The reviewers note that eukaryotes and archaea were not investigated so it is suggested that in most places, specifically abstract, material and methods, and results that the term bacterial instead of microbial be used so as not to confuse readers. This approach provides a more targeted focus to the manuscript that highlights the data shown.

Please let us deal with some inconsistencies first prior to going page by page. The sample names are not consistently labeled in the tables and figures. The reviewers prefer the listing in table 1. The reviewers prefer the FW for “fracture water” not fluids which was defined in line 134. We are confused by the statement in material and methods that one service water was sampled over an extended time frame, but that there are 3 service waters listed in supplemental table 2. Was SW 2018 the only sample used for figures 3 and 5? In Figure 4 there are two service water samples, but they are not designated by year. Why not use all 3 SW samples for all the assays? DNA recovery? It is also suggested that you provide a star designation by the biosampler name for the one exposed to mine air. This makes it easier to see and notice the differences.

General comments - page by page.

Kidd Creek Observatory “KCO” potentially use an acronym

P3 L63 Martian subsurface

P5 L121-125. Sentence is too long. Consider deleting “regardless” “investigated and”. Consider using “genomic and phospholipid investigations have not been carried out to date. PLFA not defined.

P5 L126 information on the indigenous

P5-6 L128-134 very long sentence. Consider making it two sentences.

P6 L136 KCO for observatory

P6 L138 bacterial for microbial

P7 L170 add figure 1 at end of sentence.

P7 L174-175 Three service water samples were collected.

P7 L184 -186 17.3 L of service water was collected where (at pump, in mine near borehole)? And into what type of container? Explain why you did not use a secondary 0.1um filter following the 0.22um filter to capture the smaller size bacteria often associated with deep subsurface environments. Was the 17.3 L filtered on separate filters for DNA and PLFA or was the filter split? What volume was used for each sample type?

P7 L187 Caps in Title

P8 L188 Please consider: By mass, 10% of the glass wool, sampled of multiple locations within the biosampler formed a composite sample for DNA analysis.

P7 L194 Was the glass wool also freeze-dried before the water rinse? It appears that the DNA analysis is based only on the rinse that was filtered and not the residual biofilm or glass wool. If so then that might explain some of the reduced DNA recovery. Consider providing a more detailed explanation here of how the samples were processed. A side note: would sonication have helped recovery cells from matrixes?

P8 L208 Please use “bacterial” for prokaryotic as no archaea were analyzed.

P8 L212 “each PCR was prepared in triplicate” but on L214 “Replicate PCR” – this is not clear triplicate or replicate PCR reactions were done?

P8 L215 Authors indicated that equal quantities of each amplicon were pooled, but for samples that did not yield amplicons and for controls a volume of 5 ul was added to the sequencing pool. Could authors explain why volume of 5 ul was used.

P9 L244 Were all PLFA samples freeze-dried?

P10 L256. Please include trans and cis designations.

P10 L265 Was the ¹³C isotope value of the methanol used for methylation measure? If so, what was the value?

P10 L270 Please consider “fracture” instead of borehole

P10 L271 how was PLFA normalized for statistical analyses?

P11 L278 Consider using Fracture Waters

P11 L286 Please consider: “Continuous natural flushing of both boreholes is due to artesian flow which contributes to ...”

P11 L302 Delete “quantification of”.

P11 L305. Also consider substrates – degrading cells

P12 L320 on Number of PLFA peaks or listed but not identified. The reviewers suggest the authors add text which describe the predominance of certain fatty acids in each type of sample prior to the ¹³C data.

P12 L332 “relative” abundance

P12 L336 and P13 L348 and L350. Three PLFA – please identify them by name

P12 L385 ... Double check r values are often missing a zero on the front side of decimal point.

P14 L393-394 Based on Table 1, where authors estimated cellular biomass using PLFA measurements and conversion factor, authors had enough microbial cells to get DNA and in this case, it is not competent to explain lack of success in DNA amplification by “relatively low biomass” – all samples had similar biomass. Authors pointed the high salinity as one of inhibition factor. What about high cation and anion concentration? May cation/anion that were present in the original samples be coextract with DNA and further inhibit PCR reaction? Any other potential inhibition factors should be discussed.

P15 L411 Why is there no result section detailing the metabolic profiling?

P15 L414 and L421 “bacterial” for microbial

P16 L428 “...for this component...” why bacterial microorganisms called this way? Change to member or bacterium. Then L430 change to “member of the in situ...”

P438 Space missing in front of Daly

P17 L463 First sentence had reviewer think of estimated redox values. Have those been calculated for these fracture waters? Quinone analysis would have been interesting results.

P18 L 489 “biome under investigation there [missed dot here] The authors”

P18 L491 change make them to make those bacteria

P28 L909 please superscript 18

Table 1. Please consider these suggested changes:

Sample size (g or ml) instead of mass of sample. This way we do not have to back calculated to get volume of SW filtered. Why not change Total PLFA (pmol). To pmol/g; Cells/g to Cells/g or ml. This way both the PLFA on column (which was not explained) and the Total Cells column could be deleted. This makes it easier for the reader to then use estimate the cells for the DNA samples (which the samples size was estimated at about 200mg in the supplemental material). Maybe this should be mentioned in M&M because, the reviewers thought the glass wool from end cap was included with the bulk glass wool such that it was thought that ~ 3 g of glass wool was used for DNA analysis. A note on the biofilms, it is assumed that this weight included the dried salts after sample was freeze dried. Generally, it is better to describe the biofilms as cm³. However, getting the dimensions in the field is often hard to measure and the gram weight is an acceptable alternative. Provide border line to separate FW from biofilms and biofilms from SW.

Table 2. ok

Figure 1. Needs some work. There is no scale of measure provided. The biosamplers and flow meters are not identified. Consider adding FW to the numbers since they are placed next to the tubing or move the name closer to the borehole piping.

Figure 2. First of all, this is the only place a true listing of the individual FAMES. Suggestions for improvement will be described below, but first some comments for the figure as is, which is small to read and busy. In the legend, D is a single biosampler. The vertical axis are not the same values. Figures A, C and E are averages of two samples yet there are no standard deviation/error bars.

To improve, please consider putting the individual PLFA mole percent values in a table in supplemental section. It may be worth noting that polyunsaturates (18:2, 18:3) were detected and may be indicative of eukaryotic presence. It is assumed that only the peaks with detectable ¹³C were labelled (generally not all FAMES are at a high enough concentration to get an

isotope value. Thus, it is suggest that the figures contain only those FAMES with the overlay ^{13}C value. And cite in the legend the expanded profiles in the supplemental.

Figure 3. SW date

Figure 4. Suggest using bacterial for microbial. Please bold or enlarge "ASV taxonomic affiliation and ASV phylum affiliation. Please consider adding the g, s, and f designations in the legend as genus, species and family. This helps reader who may be unfamiliar with designations. We suggest some color changes that may help increase visibility and make it easier to see difference if it is in black and white. Try to make the lighter green a lighter color or another color. How about using a lighter shade of grey or another color for the circles on the service waters (no dates). Please notice the difference in sample names.

Figure 5. Consider adding the last sentence in figure 3 to this figure. SW date

Supplemental materials:

First line:

Suggest using "Fractured waters from the two boreholes"

Do the references in this section receive a number designation as in the main text?

Third line under discussion. Suggest using "external mine environment"

With the deference in the biofilm samples, why were they combined in figure 2?

Nine lines down, page 2: Homogenizing – was this done for both DNA and lipid samples? Did you consider sonication as an alternative?

S table 1: Sample names. With the flow rates available could you not provide a column of the estimated total volumes that passed through the biosamplers.

S table 2: sample names

Reviewer #2 (Remarks to the Author):

NA Please see the attached file.

Reviewer #3 (Remarks to the Author):

Summary:

The manuscript presents an analysis of samples collected from a deep terrestrial habitat. The authors reason that *Candidatus Frackibacter* is native to the fracture water at depth. The manuscript makes a convincing case that the samples which produce the abundance of *Candidatus Frackibacter* are not contaminated. However, the case for the metabolic role and ecology of *Candidatus Frackibacter* is based on the observation of the ASV in this sample and a PLFA analysis and this is not sufficient to make a convincing case.

Specific Comments:

L83-85: This final sentence of this paragraph introduces a new detail and doesn't elaborate, what is the paragraph conclusion sentence? Please revise.

Paragraph starting at L86: This is a review of the studies conducted at Kidd Creek Observatory with no details. Can these details be folded into the introduction which should build the case for this study at this site? Otherwise, this paragraph can be incorporated into the discussion.

L124: Define PLFA on first use here (it is defined at line 139).

L146-149: This is repeated from the introduction (L93-96 and L120-123).

Site Description: What about the geology of the mine and geologic units intercepted by these studied boreholes? Some of these higher level characteristics are more appropriate here since the specific geochemistry is likely discussion material when describing the microorganisms found in these boreholes.

Since FW12261 did not yield sufficient reads, we need to know more about how these two boreholes are different. Are they right next to each other?

L205: What were the extracted DNA concentrations from the controls? Good to mention here, but do acknowledge you state the lack of a band on the gel.

L216: Amplicon, is that a gel band? "PCR product" is a better term than amplicon.

L293-295 and 306-311: This is discussion material, but how does oxygen get into the fracture was if it is openly flowing out? Assuming these are reducing fluids, there is some oxygen buffering capacity and oxygen has a hard time diffusing into flowing water and making it into the fracture water. Yes, maybe things are different right at the outlet, but I would favor the explanation that flowing water didn't allow biofilms to establish on the walls of the borehole, whereas 12299 did allow for that and then once it started flowing again, washed the biofilm and cells into your biosampler.

L364: What are these p and r values referring to?

L420: What is the DOC concentration in these waters?

Conclusion ending at L495: The evidence for this is not very compelling, especially the connection between carbon sources and *Candidatus Frackibacter* metabolism throughout the global distribution of this species. Too many leaps in logic.

Section starting at L499: See comment above regarding flowing water and oxygen. Maybe I do not understand well enough how the water flows out of the borehole.

L550: What is the “-28 per mille group”?

L559-561: Suggest revising the ending of this paragraph for clarity. Need to explain gold mine results and incomplete sentences.

L562: My big question is why doesn't this organism use the abundant and plentiful organic acids? Why is it eating, what seems to be, relict geologic organic matter. The energetics of that don't work out.

L585: Figure 4 show 3 ASVs of *Candidatus Frackibacter*, so it is more than one bacterium. There seems to be some diversity within *Candidatus Frackibacter*.

L592-594: Is the single, abundant *Sphingomonas* ASV indicative of a deep mine environment, but not a surface contaminant? There is value in that conclusion, but that doesn't come across.

L596-603: Bringing in new lines of evidence in the conclusion section adds confusion, suggest revising to focus on these results.

General Comments:

Paragraph beginning at L126: The motivation, as written, for this study seems to be new methods applied to unique samples. What was the hypothesis and motivation for designing experimental work and studying these samples?

Results Section: Several of the results paragraphs have discussion-like comparisons to other studies and sometimes the results aren't presented at all (i.e., L279-283). The reader needs to know what was measured in this study.

Discussion Section: The repetition through the first four discussion paragraphs without presenting new lines of evidence to support the contention of *Candidatus Frackibacter* being native to fracture water needs to be revised.

Focus on two ASVs in Discussion: The Discussion largely focuses on two ASVs of the ~400 found. Given this deep dive into these two ASVs, their metabolism, global distribution, ecology, etc., it needs a different method. I hate suggesting “deeper sequencing” because I work in low biomass environments and low volume samples, but it would seem to get at your question more directly. If these two ASVs are so abundant, a metagenome should produce good metagenome assembled genomes. This would add some weight to those discussions surrounding the two ASVs and their metabolism (i.e., L552). Ultimately, this gets back to my first general comment, what was the hypothesis or question that motivated this work? It appears to be studying the ecological role of “indicator” or abundant microbial species in these two habitats. 16S amplicon sequencing is not the appropriate method to get at this question. It is for a survey of the entire community, but that is not what is presented in this paper.

Overall, the paper does a good job of demonstrating the 12299 fracture water and biofilm samples are not contaminated.

Communications Earth & Environment is committed to improving transparency in authorship. As part of our efforts in this direction, we are now requesting that all authors identified as ‘corresponding author’ create and link their Open Researcher and Contributor Identifier (ORCID) with their account on the Manuscript Tracking System prior to acceptance. ORCID helps the scientific community achieve unambiguous attribution of all scholarly contributions. You can create and link your ORCID from the home page of the Manuscript Tracking System by clicking on ‘Modify my Springer Nature account’ and following the instructions in the link below. Please also inform all co-authors that they can add their ORCIDs to their accounts and that they must do so prior to acceptance.

Version 1:

Decision Letter:

Dear Dr Slater,

Your manuscript titled "Deep terrestrial indigenous microbial community dominated by *Candidatus Frackibacter*" has now been seen by our reviewers, whose comments appear below (Reviewer #2 informed us that they are satisfied with the revisions). In light of their advice we are delighted to say that we are happy, in principle, to publish a suitably revised version

in Communications Earth & Environment.

We therefore invite you to revise your paper one last time to address the remaining concerns of our reviewers. At the same time we ask that you edit your manuscript to comply with our format requirements and to maximise the accessibility and therefore the impact of your work.

EDITORIAL REQUESTS:

****Please take care to match our formatting and policy requirements. We will check revised manuscript and return manuscripts that do not comply. Such requests will lead to delays. ****

SUBMISSION INFORMATION:

OPEN ACCESS:

Communications Earth & Environment is a fully open access journal. Articles are made freely accessible on publication. For further information about article processing charges, open access funding, and advice and support from Nature Research, please visit <https://www.nature.com/commsenv/open-access>

Link Redacted

Best regards,

D'Arcy Meyer-Dombard, PhD
Editorial Board Member
Communications Earth & Environment
orcid.org/0000-0001-9862-4839

Joe Aslin
Deputy Editor,
Communications Earth & Environment
<https://www.nature.com/commsenv/>
Twitter: @CommsEarth

REVIEWERS' COMMENTS:

Reviewer #1 (Remarks to the Author):

This paper describes the deep and isolated microbial communities in fracture water in the Kidd Creek Observatory. The authors compare the dominance of *Candidatus Frackibacter* in isolated fracture water to the dominance of *Sphingomonas* in fracture water exposed to mine environment. Cross disciplinary methods of geochemistry, lipid and DNA analyses were used to support the findings.

This revised and resubmitted paper is well written and clearly discusses the major findings. The authors appropriately

addressed reviewers comments and it shows in the quality of this version. Great job.

Reviewer #2 (Remarks to the Author):

NA

Reviewer #3 (Remarks to the Author):

The authors have made many changes that have improved the manuscript. Below are some specific comments.

Introduction Paragraph 2: The case for Mars is made well, but not the other satellites. Those should be removed because their case hasn't been made and they are ocean worlds. Does this come back around in the discussion?

L155: Not statistically different, so $p > 0.01$, right? Depends on how you stated your hypothesis and set up the test. A few lines down waters are significantly different using the same test and p is still < 0.01 .

L158-161: Something is confusing here. A few sentences ago the waters from 12299 and 12261 were statistically identical and now one is more representative than the other. Please revise.

L237: ASVs are usually number ordered by their abundance in the batch of samples processed. If all these samples were processed together (FW and SW), then who are ASVs 1-13 because those are high abundance ASVs. Are you referencing the Supplemental ASV table at line 244? I don't understand what is presented in that file.

Results: A rank-order list of the top 20 most abundant ASVs and their relative abundances from the different samples would really help to contextualize how abundant these ASVs are relative to everything else that was found in these samples. Like Fig. 4, but on Fig. 4 it is not clear if these are the top 26 most abundant ASVs from the entire dataset (the ASV numbering would imply not).

L283: I understand FAPROTAX is a published method, but the organism in question is a Candidatus Frackibacter. How well does FAPROTAX deal with uncultured and undescribed species?

L356: The comparison to food fermentation factories needs to be described better. What does this tell us about your study?

L495: "producing delivering" please revise.

L510-522: I still find this discussion of the metabolic potential speculative.

Conclusion section: Can you provide a 2-3 sentence conclusion of the stable isotope analysis? That would help tie up those data.

Results/Figure 5. I don't fully understand what is being correlate and how it matters

Review comments for “Deep terrestrial saline fluids harbor an indigenous microbial community dominated by *Candidatus Frackibacter*”

In the manuscript by Ford et al., the researchers studied the microbial communities within the Kidd Creek Observatory subsurface fracture water system with mean residence time of 25 hundreds of millions to over one billion years. Based on amplicon sequencing and stable carbon isotope analyses, it was shown that *Candidatus Frackibacter* of family *Halobacteroidaceae* prevailed in the deep subsurface and used the carbon derived from ancient carbon-rich layers common in the systems. Thus, the researchers proposed that the dominating *Candidatus Frackibacter* species were indigenous in the ancient deep subsurface environment. This work is interesting and complementary for the earlier discovery of *Candidatus Frackibacter* in the geological formation for fractured shale gas. The results also suggest the life in other planets. The topic is interesting and of fundamental significance. The paper was well written and easy to follow.

In the study, the researchers carefully set up a series of controls to ensure quality of the analyses. The project itself is very challenging considering the low biomass in the deep subsurface and potential exogenous contamination. The major limitation of the work was that the genomic DNA concentration was very low so that the amplicon under sequencing was not visible in the agarose gel for some samples. This is not common for amplicon analyses to use the samples invisible even for the PCR products. The low biomass also led to constrained amplicon and isotope analyses. Considering the objective of the work, the available evidence could have been stronger.

There are a few specific comments for the manuscript:

Lines 98-99. It is common to observe elevated pH during serpentinization. Did mineralogical characterization support serpentinization? The pH in the results, however, showed that it was weakly acidic, which is not consistent to most serpentinizing environments. Were the C-containing compounds and H₂ solely from abiotic reactions? How about microbial processes? How about other geochemical factors, such as temperature, salinity, conductivity?

Lines 188-190. How about the microbial distribution at different locations in the study? Would composite sample development create biased results?

Lines 190-193. The researchers used PCR-grade water to remove salt from the samples prior to DNA extraction. Would such operation result in lysis of some cells, especially for those

from saline environment?

Line 236. What was the reference for Decontam? Please provide.

Line 325. “Service” was redundant. What was service water?

Line 365. What was the sequencing depth for the amplicons? How many reads were generated for each sample?

Lines 416. How was chemoheterotrophic metabolism inferred? What can act as the potential nutrient source for the predicted lifestyle?

Line 460. The discussion of the metabolic potential of the *Candidatus Frackibacter* in the study was mainly based on the earlier studies. It would have been much better to provide other lines of evidence to support consumption of geogenic organic nutrients by the *Candidatus Frackibacter*. This is crucial for the conclusion that the *Candidatus Frackibacter* species are indigenous and can rely on the available energy and nutrient source for survival.

Lines 488-490. Please double check this sentence.

Line 492. It was not very clear about carbon bioavailability for the carbon-rich host rocks, which would significantly influence the availability of organic nutrients for the native microorganisms.

Other, the discussion in SI “*Comparison with Previous Analysis by Co-authors*” can be moved to the main text. The information of potential metabolism of *Candidatus Frackibacter* to use sulfate or iron for energy generation can be integrated with the geochemistry of the samples under consideration to support the microbial lifestyle in deep subsurface.

Manuscript COMMSENV-24-0835-T

Dear Dr. D'Arcy Meyer-Dombard and Joe Aslin,

Enclosed is our detailed response to the all reviewer feedback in blue, with line numbers where applicable referring to the tracked changes document, on the manuscript COMMSENV-24-0835-T:

Deep terrestrial indigenous microbial community dominated by *Candidatus Frackibacter*

by Sian E. Ford¹, Greg F. Slater¹, Katja Engel², Oliver Warr^{3,4}, Garnet S. Lollar³, Allyson Brady^{1,5}, Josh D. Neufeld³, Barbara Sherwood Lollar^{2,6}

Response to Reviewers

Reviewer #1

This research is novel and extends the information on isolated highly saline, deep subsurface bacterial communities in the Kidd Creek Observatory. Comparison of the more anaerobic fracture waters with biofilm and service water samples helps distinguish these isolated bacterial communities and relate their DNA and PLFA compositions to carbon sources and metabolic processes. The paper is well written but there is room for improvement and clarification in some areas prior to publication.

We appreciate the positive comments of this reviewer and address their points of clarification and improvement in detail below.

Please note that “indigenous” is misspelled on cover page.

Thanks for spotting this typo, which is now corrected.

P2 line 24, We suggest switching microbial to bacterial. And why:

The reviewers note that eukaryotes and archaea were not investigated so it is suggested that in most places, specifically abstract, material and methods, and results that the term bacterial instead of microbial be used so as not to confuses readers. This approach provides a more targeted focus to the manuscript that highlights the data shown.

The primers used in our analyses are universal, targeting both bacteria and archaea. Because we detect archaea in our data, as presented in Figure 4, it is correct to say “bacteria and archaea” or “microorganisms” rather than “bacteria” alone. Likewise, phospholipid fatty acid analysis encompasses detection of both bacteria and eukaryotes.

Please let us deal with some inconsistencies first prior to going page by page. The sample names are not consistently labeled in the tables and figures. The reviewers prefer the listing in table 1. The reviewers prefer the FW for “fracture water” not fluids which was defined in line 134.

The suggested change has been made throughout the manuscript and sample names are now consistent.

We are confused by the statement in material and methods that one service water was sampled over an extended time frame, but that there are 3 service waters listed in supplemental table 2.

We have clarified the sampling and analysis of service water on Line 698: “Service water was monitored for geochemistry at all sampling points (Supplementary Table S2). Only the January 29, 2018, service water sample was used for PLFA and DNA analysis”.

Was SW 2018 the only sample used for figures 3 and 5?

This has now been clarified as per the previous reviewer comment (See Line 698), wherein it is made clear that only the SW 2018 sample was used for PLFA and DNA.

In Figure 4 there are two service water samples, but they are not designated by year.

As with the preceding two comments, this has been clarified in the Methods section on Line 698. Additionally, it has been made clear in the Methods at line 708 and in the caption for Figure 4 at Line 1563 that the service water sample from 2018 was run in duplicate.

Why not use all 3 SW samples for all the assays? DNA recovery?

Only the 2018 Service Water sample was collected specifically for DNA and PLFA analysis. The SW samples from previous years were only taken for geochemical monitoring and as such were not in large enough quantities for PLFA and DNA analysis. This has now been clarified on Line 698.

It is also suggested that you provide a star designation by the biosampler name for the one exposed to mine air. This makes it easier to see and notice the differences.

We appreciate this suggestion. The star designation for this sample has been added to the text and all figures and accompanying captions as *FW12261-2018.

General comments - page by page.

Kidd Creek Observatory “KCO” potentially use an acronym

The suggested change has been made throughout.

P3 L63 Martian subsurface

Change made as requested.

P5 L121-125. Sentence is too long. Consider deleting “regardless” “investigated and”. Consider using “genomic and phospholipid investigations have not been carried out to date. PLFA not defined.

This sentence has now been reworded into two sentences at Line 123.

P5 L126 information on the indigenous

Unfortunately, the reviewer’s comment appears to have been cut off mid-sentence, however we have assumed the reviewer is requesting more information on what has been found previously. This information has been added in to the discussion beginning at Line 505.

P5-6 L128-134 very long sentence. Consider making it two sentences.

This sentence has been split into two as requested.

P6 L136 KCO for observatory

This change has been made as requested.

P6 L138 bacterial for microbial

As described above, our primers are not exclusive to bacteria and we have kept “microbial” here.

P7 L170 add figure 1 at end of sentence.

Change made as suggested.

P7 L174-175 Three service water samples were collected.

This has been clarified as described above.

P7 L184 -186 17.3 L of service water was collected where (at pump, in mine near borehole)? And into what type of container? Explain why you did not use a secondary 0.1um filter following the 0.22um filter to capture the smaller size bacteria often associated with deep subsurface environments. Was the 17.3 L filtered on separate filters for DNA and PLFA or was the filter split? What volume was used for each sample type?

The protocol for the service water sample has been expanded in the Methods section to increase clarity for the reader at Lines 698, 708-714, 723. Regarding filters, this study was designed in line with the preceding microbiology studies at Kidd Creek by Lollar et al. [42] and Wilpiseski et al. [44], which likewise used 0.22 µm filters. That said, the inclusion of a 0.1 µm filter is a good suggestion and will be applied for future sampling campaigns.

P7 L187 Caps in Title

This error is now corrected.

P8 L188 Please consider: By mass, 10% of the glass wool, sampled of multiple locations within the biosampler formed a composite sample for DNA analysis.

This change was made as suggested.

P7 L194 Was the glass wool also freeze-dried before the water rinse? It appears that the DNA analysis is based only on the rinse that was filtered and not the residual biofilm or glass wool. If so then that might explain some of the reduced DNA recovery. Consider providing a more detailed explanation here of how the samples were processed. A side note: would sonication have helped recovery cells from matrixes?

We appreciate the suggestion that we expand on our methods here, which we now outline in more detail at Lines 716-725 and 740. In particular, we should clarify that the glass wool was freeze-dried prior to water rinsing, and we indeed used sonication to maximize cell recovery.

P8 L208 Please use “bacterial” for prokaryotic as no archaea were analyzed.

As per our previous responses, the primers used are universal, targeting both archaea and bacteria, while PLFA analysis surveys both bacteria and eukarya. Therefore, it is correct to say “prokaryotic” or “microorganisms” rather than “bacteria” alone, and so this original wording has been preserved within the manuscript.

P8 L212 “each PCR was prepared in triplicate” but on L214 “Replicate PCR” – this is not clear triplicate or replicate PCR reactions were done?

We have made a change to clarify that triplicate analysis was conducted.

P8 L215 Authors indicated that equal quantities of each amplicon were pooled, but for samples that did not yield amplicons and for controls a volume of 5 ul was added to the sequencing pool. Could authors explain why volume of 5 ul was used.

If no amplicon was detected in the agarose gel, no (or very low) Illumina 16S rRNA gene sequence results are expected. In a routine 16S rRNA gene sequencing pipeline, samples that do not have an amplicon would not be included in the Illumina 16S rRNA gene sequencing pool.

We do include those samples at a low volume anyway because often these samples yield a few reads, allowing us to identify contaminant sequences in no-template controls (NTCs) and sometimes even see sample-specific signals. We choose to pool only 5 μ l volumes of sample without amplicon because adding too much volume for these reduces/dilutes the concentration of template in the Illumina pool and can interfere with downstream processing.

P9 L244 Were all PLFA samples freeze-dried?

We revised this sentence to clarify at Line 779 that all PLFA samples were freeze-dried.

P10 L256. Please include trans and cis designations.

These designations were added as requested at Line 790.

P10 L265 Was the ^{13}C isotope value of the methanol used for methylation measure? If so, what was the value?

The $\delta^{13}\text{C}$ of the methanol used for methanolysis was used for the methylation measure and had a value of -37.4‰. We have added these details to the text at Line 797.

P10 L270 Please consider “fracture” instead of borehole

This change was made as requested.

P10 L271 how was PLFA normalized for statistical analyses?

The PLFA data were sum normalized and this is now clarified in the text on Line 809.

P11 L278 Consider using Fracture Waters

This change was made as suggested.

P11 L286 Please consider: “Continuous natural flushing of both boreholes is due to artesian flow which contributes to ...”

This change was made as suggested on Line 158.

P11 L302 Delete “quantification of”.

This change was made as suggested.

P11 L305. Also consider substrates – degrading cells

We recognize in the manuscript at Lines 345 and 685 that the biosampler material likely provides a surface on which microorganisms can attach and grow, and we argue that this growth must be supported by the incoming water and gas. While degrading cells may provide a substrate, the biomass of the fluids is very low and so the contribution of this would be lower still. Furthermore, any recycling of degrading cells must still be supported by the same initial geochemical energy source, and so are not a distinct support for the microbial community.

P12 L320 on Number of PLFA peaks or listed but not identified. The reviewers suggest the authors add text which describe the predominance of certain fatty acids in each type of sample prior to the ^{13}C data.

The requested data have been added for each sample in higher classification in a new Figure 2, and with more specificity, briefly in the Results section, and in Supplemental Table S3.

P12 L332 “relative” abundance

This change was made as suggested at Line 192.

P12 L336 and P13 L348 and L350. Three PLFA – please identify them by name

The PLFA names have been added as requested.

P12 L385 ... Double check r values are often missing a zero on the front side of decimal point.

This correction has been made as suggested.

P14 L393-394 Based on Table 1, where authors estimated cellular biomass using PLFA measurements and conversion factor, authors had enough microbial cells to get DNA and in this case, it is not competent to explain lack of success in DNA amplification by “relatively low biomass” – all samples had similar biomass. Authors pointed the high salinity as one of inhibition factor. What about high cation and anion concentration? May cation/anion that were present in the original samples be coextract with DNA and further inhibit PCR reaction? Any other potential inhibition factors should be discussed.

Both the high salt concentrations in the samples and cell lysis caused by freeze-drying likely influenced DNA recovery. The washing step to eliminate salt would have also likewise eliminated inhibiting cations and/or anions. We appreciate these points raised by the reviewer and have therefore added further description of sample collection and processing has been added at Lines 664, 689-725, and 740.

P15 L411 Why is there no result section detailing the metabolic profiling?

Previously this information had been folded into the discussion. However, to avoid confusion, these data have been added to the Results section with the accompanying figure (now Figure 6) that was previously in the supplemental information.

P15 L414 and L421 “bacterial“ for microbial

As per previous comments, the primers used in our DNA analysis were universal and so microbial is the most accurate term to use here and elsewhere.

P16 L428 “...for this component...” why bacterial microorganisms called this way? Change to member or bacterium. Then L430 change to “member of the in situ...”

The first change has been implemented at Line 360, and the second sentence now rewritten for clarity.

P438 Space missing in front of Daly

This correction was made.

P17 L463 First sentence had reviewer think of estimated redox values. Have those been calculated for these fracture waters? Quinone analysis would have been interesting results.

Estimated redox values have not been calculated for these fracture waters. While we agree that oxidation-reduction potential data as well as quinone analysis would be interesting we do not have them available.

P18 L 489 “biome under investigation there [missed dot here] The authors”

Change made as suggested.

P18 L491 change make them to make those bacteria

Change made as suggested.

P28 L909 please superscript 18

Change made as requested.

Table 1. Please consider these suggested changes:

Sample size (g or ml) instead of mass of sample. This way we do not have to back calculated to get volume of SW filtered.

Why not change Total PLFA (pmol). To pmol/g; Cells/g to Cells/g or ml. This way both the PLFA on column (which was not explained) and the Total Cells column could be deleted. This makes

it easier for the reader to then use estimate the cells for the DNA samples (which the samples size was estimated at about 200mg in the supplemental material). Maybe this should be mentioned in M&M because, the reviewers thought the glass wool from end cap was included with the bulk glass wool such that it was thought that ~ 3 g of glass wool was used for DNA analysis.

The requested changes have been made as suggested. In Table 1 on Page 45 the first column is now Sample size (g or mL). The total PLFA has been changed to PLFA abundance and Cellular abundance. Towards the latter suggestion changes have been made in the Methods section to clarify how our analysis was done, as per our previous responses.

A note on the biofilms, it is assumed that this weight included the dried salts after sample was freeze dried. Generally, it is better to describe the biofilms as cm³. However, getting the dimensions in the field is often hard to measure and the gram weight is an acceptable alternative.

The reviewer raises a good point that we do not have volume measurements for the biofilms and so, as highlighted, mass was used instead.

Provide border line to separate FW from biofilms and biofilms from SW.

This change was made as suggested in all tables beginning at Page 45.

Figure 1. Needs some work. There is no scale of measure provided. The biosamplers and flow meters are not identified. Consider adding FW to the numbers since they are placed next to the tubing or move the name closer to the borehole piping.

Thank you for these suggestions and indeed this figure has been improved and it has now been revised (Page 49, Line 1530). Borehole labels have been moved adjacent to the boreholes. Flowmeters and biosamplers are labeled. An approximate scale has also been added.

Figure 2. First of all, this is the only place a true listing of the individual FAMES. Suggestions for improvement will be described below, but first some comments for the figure as is, which is small to read and busy. In the legend, D is a single biosampler. The vertical axis are not the same values. Figures A, C and E are averages of two samples yet there are no standard deviation/error bars.

To improve, please consider putting the individual PLFA mole percent values in a table in supplemental section. It may be worth noting that polyunsaturates (18:2, 18:3) were detected and may be indicative of eukaryotic presence. It is assumed that only the peaks with detectable ¹³C were labelled (generally not all FAMES are at a high enough concentration to get an isotope value. Thus, it is suggest that the figures contain only those FAMES with the overlay ¹³C value. And cite in the legend the expanded profiles in the supplemental.

Thank you. The most abundant PLFA/FAMEs are now listed within the text in the relevant results section beginning at Line 198. Figure 2 has been replaced with a stacked bar graph on Page 50, which now highlights relative abundances of PLFA classes. The stable carbon data has been prepared as a table (Table 2, Page 46). As suggested, a supplemental table containing the relative mole abundance of all PLFA has been prepared and included as Supplemental Table S3.

Figure 3. SW date

The service water date has been added as requested.

Figure 4. Suggest using bacterial for microbial. Please bold or enlarge “ASV taxonomic affiliation and ASV phylum affiliation. Please consider adding the g, s, and f designations in the legend as genus, species and family. This helps reader who may be unfamiliar with designations. We suggest some color changes that may help increase visibility and make it easier to see difference if it is in black and white. Try to make the lighter green a lighter color or another color. How about using a lighter shade of grey or another color for the circles on the service waters (no dates). Please notice the difference in sample names.

We have applied several edits to Figure 4 including adding the legend for g, s, and f. As per our previous response, sample names have been made consistent with the date, 2018, added for the service water sample. The colors have been adjusted to be more greyscale friendly.

Figure 5. Consider adding the last sentence in figure 3 to this figure. SW date

Service water date added. Last sentence from Figure 3 added to Figure 5.

Supplemental materials:

Please note all supplemental information has been incorporated into the main text, with all suggested changes made.

First line:

Suggest using “Fractured waters from the two boreholes”

Change made as suggested, with some rewriting to fix run-on sentences.

Do the references in this section receive a number designation as in the main text?

Although they did, the supplemental writing has been incorporated into the main text and the supplemental references have been moved accordingly.

Third line under discussion. Suggest using “external mine environment”

This change has been made as suggested.

With the deference in the biofilm samples, why were they combined in figure 2?

This comment is well taken and, due to these differences, the biofilm samples have now been separated by year as Biofilm 2016 and Biofilm 2017 in all figures and tables.

Nine lines down, page 2: Homogenizing – was this done for both DNA and lipid samples? Did you consider sonication as an alternative?

Homogenization and sonication were done for all samples for both DNA and PLFA analysis. This is now clarified in the main text on Line 721.

S table 1: Sample names. With the flow rates available could you not provide a column of the estimated total volumes that passed through the biosamplers.

Unfortunately, due to variations in flow rate that occur over time, we have low confidence in our ability to provide a reasonable total. Further, due to the long deployment and potential for growth in the samplers that we cannot constrain, we do not feel that the total flow can be effectively compared to the biomass detected.

Sample names have been made consistent.

S table 2: sample names

Sample names have been made consistent.

Reviewer #2

In the manuscript by Ford et al., the researchers studied the microbial communities within the Kidd Creek Observatory subsurface fracture water system with mean residence time of 25 hundreds of millions to over one billion years. Based on amplicon sequencing and stable carbon isotope analyses, it was shown that *Candidatus Frackibacter* of family Halobacteroidaceae prevailed in the deep subsurface and used the carbon derived from ancient carbon-rich layers common in the systems. Thus, the researchers proposed that the dominating *Candidatus Frackibacter* species were indigenous in the ancient deep subsurface environment. This work is interesting and complementary for the earlier discovery of *Candidatus Frackibacter* in the geological formation for fractured shale gas. The results also suggest the life in other planets. The topic is interesting and of fundamental significance. The paper was well written and easy to follow.

We appreciate the positive comments of this reviewer.

In the study, the researchers carefully set up a series of controls to ensure quality of the analyses. The project itself is very challenging considering the low biomass in the deep subsurface and potential exogenous contamination. The major limitation of the work was that the genomic DNA concentration was very low so that the amplicon under sequencing was not visible in the agarose gel for some samples. This is not common for amplicon analyses to use the samples invisible even for the PCR products. The low biomass also led to constrained amplicon and isotope analyses. Considering the objective of the work, the available evidence could have been stronger.

As mentioned throughout the manuscript, there were two samples (FW12261-2017A and FW12261-2017B) that did not yield a visible amplicon and only resulted in very few sequences, thus we did not present those results. To be clear, the other samples we analyzed and presented in the manuscript all had visible amplicon bands on an agarose gel and generated tens of thousands of sequences each and were distinct from no-template controls based on a Decontam analysis.

There are a few specific comments for the manuscript:

Lines 98-99. It is common to observe elevated pH during serpentinization. Did mineralogical characterization support serpentinization? The pH in the results, however, showed that it was weakly acidic, which is not consistent to most serpentinizing environments. Were the C-containing compounds and H₂ solely from abiotic reactions? How about microbial processes? How about other geochemical factors, such as temperature, salinity, conductivity?

Although it is true that serpentinization may be associated with elevated pH, the abundant H₂ gas produced within the fracture fluids can result in increased hydrogen ion concentrations and thus decrease the pH. The putative dominant abiotic production of acetate, formate, methane, and H₂ in the Kidd Creek fracture system (with potential minor microbial additions) have been extensively investigated in previous studies (e.g. refs 25, 28, 31, 54-56, 65-68). See Lines 99-106 of the revised manuscript for section discussing this characteristic at this site of predominantly abiotic signatures but low level or in some cases cryptic (e.g. methane) biological cycling.

Lines 188-190. How about the microbial distribution at different locations in the study?

This study was only able to report data from the two boreholes that were monitored: 12261 and 12299. We report on the distribution in both borehole samplers as well as the associated biofilm. We agree that future research should examine additional locations and depths to help further generalize the findings we report here.

Would composite sample development create biased results?

The biosampler is composed of a relatively large mass (~21 g) of carbon-free glass wool coated with aluminum oxide (described in the manuscript at Line 677). The length of time of

deployment may result in growth on the sampler as well as filtration as noted in Line 682. The purpose of sampling along the biosampler's length was to avoid selectively sampling only one section, which might result in missing differences in the low-abundance microbial community. The composite sample therefore provides an optimized methodology encompassing any heterogeneity that would be magnified in this low-biomass setting.

Lines 190-193. The researchers used PCR-grade water to remove salt from the samples prior to DNA extraction. Would such operation result in lysis of some cells, especially for those from saline environment?

Samples were freeze-dried prior to the salt removal with PCR-grade water, which has been clarified on Line 718 and 720. Freeze-drying offers protection against cell lysis following rehydration as the cell membrane remains in a hydrated, gel-like state. While the cells are no longer viable following freeze-drying, they remain intact and so their genetic contents protected.

Line 236. What was the reference for Decontam? Please provide.

The citation for Decontam was present on line 769 and the reference was already included in the reference section.

Line 325. "Service" was redundant. What was service water?

The redundant word was removed, and the service water description is now included on Line 707.

Line 365. What was the sequencing depth for the amplicons? How many reads were generated for each sample?

All samples presented generated tens of thousands of ASVs. The requested information has now been added to the Results section of the revised manuscript.

Lines 416. How was chemoheterotrophic metabolism inferred? What can act as the potential nutrient source for the predicted lifestyle?

The inference of chemoheterotrophy was made based on FAPROTAX results, which is now further described at Line 313. The potential nutrient sources are the focus of the discussion beginning at Line 475.

*Line 460. The discussion of the metabolic potential of the Candidatus Frackibacter in the study was mainly based on the earlier studies. It would have been much better to provide other lines of evidence to support consumption of geogenic organic nutrients by the Candidatus Frackibacter. This is crucial for the conclusion that the Candidatus Frackibacter species are indigenous and can rely on the available energy and nutrient source for survival.

As per the preceding comment, the discussion of metabolism and the carbon/nutrient source for *Candidatus Frackibacter* has been expanded to provide further lines of evidence for the consumption of geogenic organic carbon. In particular, our discussion of the isotopic compositions of the PLFA from the samples dominated by *Candidatus Frackibacter* supports our interpretation of the geogenic carbon as the primary carbon source. The isotopic signatures observed between the two groups of samples were robust, and we feel that they allowed an effective comparison to the isotopic compositions of the available carbon sources. The isotopic results strongly indicate that the organisms are not using methane or dissolved organic matter such as acetate or formate as carbon sources. Anaerobic methane oxidation is carried out by relatively few archaea, and while acetate may seem to be an ideal carbon source, not all organisms are acetotrophic. *Candidatus Frackibacter* has previously been proposed to use components of fracking fluid but detailed metabolic capabilities of this organism remain unconstrained. We feel that the evidence we have presented is effective in arguing for the proposed use of the geologically derived carbon. The similarity in ^{13}C signatures between geologic carbon in Precambrian host rocks, and the dissolved organic matter (both pools close to -27‰) has been raised frequently in past literature to support derivation of dissolved organic carbon compounds from the host rocks (refs 97, 98, 107). Hence it is not a huge step to suggest indigenous microbes may be using such dissolved species, even if they represent a small proportion of the available DOM pool at KCO compared to acetate and formate. Indeed, the use of such carbon sources has been demonstrated previously (refs 100 and 101) and the use of it by a subsurface organism is an important expansion of this potential pathway of carbon cycling that may support subsurface microbial communities.

Lines 488-490. Please double check this sentence.

This sentence has been checked and amended.

Line 492. It was not very clear about carbon bioavailability for the carbon-rich host rocks, which would significantly influence the availability of organic nutrients for the native microorganisms.

As with our previous responses, the bioavailability of the carbon-rich rock is now discussed in detail in the revised manuscript.

Other, the discussion in SI “Comparison with Previous Analysis by Co-authors” can be moved to the main text.

The information of potential metabolism of *Candidatus Frackibacter* to use sulfate or iron for energy generation can be integrated with the geochemistry of the samples under consideration to support the microbial lifestyle in deep subsurface.

We appreciate that the reviewer considers these two sections to be of sufficient interest for inclusion in the main text. As such, they have been integrated into the revised manuscript as suggested beginning at Line 598.

Reviewer #3

The manuscript presents an analysis of samples collected from a deep terrestrial habitat. The authors reason that *Candidatus Frackibacter* is native to the fracture water at depth. The manuscript makes a convincing case that the samples which produce the abundance of *Candidatus Frackibacter* are not contaminated. However, the case for the metabolic role and ecology of *Candidatus Frackibacter* is based on the observation of the ASV in this sample and a PLFA analysis and this is not sufficient to make a convincing case.

We thank the Reviewer for their support of our primary conclusion that *Candidatus Frackibacter* is indigenous to deep fracture networks. Regarding the role and specific metabolic activity that it plays here, this is a key point that was similarly raised by Reviewer 2. To address this, the discussion of metabolism and the carbon/nutrient source for *Candidatus Frackibacter* has been expanded to provide further lines of evidence for the consumption of geogenic organic carbon.

Specific Comments:

L83-85: This final sentence of this paragraph introduces a new detail and doesn't elaborate, what is the paragraph conclusion sentence? Please revise.

A sentence has been added to the end of this paragraph as suggested to provide the context and transition to the next paragraph (Line 83).

Paragraph starting at L86: This is a review of the studies conducted at Kidd Creek Observatory with no details. Can these details be folded into the introduction which should build the case for this study at this site? Otherwise, this paragraph can be incorporated into the discussion.

We have followed this suggestion to fold in information by adding a brief description of the Kidd Creek geology on Line 92. In addition, we have included discussion of the previous results at Kidd Creek within the discussion. Beginning on line 490, the isotopic compositions and composition of DOC are discussed. There is discussion of DOC release starting on line 565. And previous microbial results are discussed beginning on line 598.

L124: Define PLFA on first use here (it is defined at line 139).

This change was made as requested.

L146-149: This is repeated from the introduction (L93-96 and L120-123).

The repetitive information in these two sections is provided to contextualize the statements that follow.

Site Description: What about the geology of the mine and geologic units intercepted by these studied boreholes? Some of these higher level characteristics are more appropriate here since

the specific geochemistry is likely discussion material when describing the microorganisms found in these boreholes.

We have added a section describing the geology of the Kidd Creek system on line 92.

Since FW12261 did not yield sufficient reads, we need to know more about how these two boreholes are different. Are they right next to each other?

Further description of the statistical analysis of the geochemistry of the fracture waters from the two boreholes has been added at Line 811. Additionally, further description of the two boreholes has been expanded upon at Line 664, and a scale has been added to the revised Figure 1 on Page 49.

L205: What were the extracted DNA concentrations from the controls? Good to mention here, but do acknowledge you state the lack of a band on the gel.

The requested information has been added at Line 740: "DNA concentrations of controls were below the detection limit of the Qubit dsDNA High Sensitivity Assay Kit (Invitrogen)."

L216: Amplicon, is that a gel band? "PCR product" is a better term than amplicon.

Yes, an amplicon is a PCR product and both words can be used synonymously. We have changed the wording as suggested.

L293-295 and 306-311: This is discussion material, but how does oxygen get into the fracture was if it is openly flowing out? Assuming these are reducing fluids, there is some oxygen buffering capacity and oxygen has a hard time diffusing into flowing water and making it into the fracture water. Yes, maybe things are different right at the outlet, but I would favor the explanation that flowing water didn't allow biofilms to establish on the walls of the borehole, whereas 12299 did allow for that and then once it started flowing again, washed the biofilm and cells into your biosampler.

The more discussion-like material has been moved to the relevant section. To clarify the orientation of the boreholes and biofilms, Figure 1 has been revised (Page 49) and we hope that this helps clear up some confusion. We agree that at the outlet of the borehole specifically that there may be oxygen present, particularly when the variable flow of water and gas is lower. Further to this point, the supplemental discussion that addresses this intermediate state has been integrated into the revised manuscript beginning on Line 453.

L364: What are these p and r values referring to?

Further description of Spearman's r has been added to the Methods section at Line 811.

L420: What is the DOC concentration in these waters?

The requested information has been added at Line 496: “The DOC concentrations have been previously reported for 12261 and 12299 as 2400 μM and 5000 μM , respectively, of which up to 90% was acetate and formate [24].”

Conclusion ending at L495: The evidence for this is not very compelling, especially the connection between carbon sources and *Candidatus Frackibacter* metabolism throughout the global distribution of this species. Too many leaps in logic.

Thank you. To address this, we have added an expanded discussion and additional references beginning at Line 519 through to Line 583, and further at Line 598 to 617.

Section starting at L499: See comment above regarding flowing water and oxygen. Maybe I do not understand well enough how the water flows out of the borehole.

Please see the response to the comment above (Lines 664 and 811, Figure 1 Page 49) clarifying this.

L550: What is the “-28 per mille group”?

We have adjusted the text to be more specific to avoid confusion introduced by using “groups.”

L559-561: Suggest revising the ending of this paragraph for clarity. Need to explain gold mine results and incomplete sentences.

This paragraph has been revised with additions to clarify results and remedy fragment sentences. Additionally, please see the comment below RE L562 for a more detailed response to the related comments regarding the discussion and major conclusions.

L562: My big question is why doesn't this organism use the abundant and plentiful organic acids? Why is it eating, what seems to be, relict geologic organic matter. The energetics of that don't work out.

*This is an important point raised by our study, and also by Reviewer #2 regarding Line 460 – see * above for the response to it there as well as the comments here. These questions are well received and because these considerations are important, discussion has been added to the revised manuscript beginning at Line 541. In addition, we have added more details at Line 519 describing the lack of evidence from previous work for microbial use of the acetate and formate pools at this site [24].

With respect to the energetics point, this certainly a question worth consideration. However, the lack of utilization of methane or acetate is consistent with the fact that not all organisms can utilize these substrates. Anaerobic methanotrophy can only be carried out by a limited number of archaea (Chadwick et al. 2022), in conjunction with only a limited number of sulphate

reducers (Murali et al. 2023). Similarly, complete oxidation of acetate cannot be achieved by all microorganisms, particularly acetogens. The energetics of oxidation beyond acetate are unfavorable (Thauer et al. 1977), and studies of acetate oxidation by Dolfing et al. (2001) demonstrated that under methanogenic conditions, the acetogenic oxidation of organic substrates is favored. Thus, as an acetogen, *Candidatus Frackibacter* may be the same as other fermentative acetogens and/or chemoheterotrophs like members of *Desulfovibrio* which cannot completely oxidize acetate (Galushko & Kuever 2019).

In extreme environments such as the waters of KCO, microbial communities can exist with an abundance of electron donors and acceptors that, while theoretically energetically viable, are either not accessed at all or at a cryptic rate that does not affect the large concentrations of these compounds in the surrounding environment [31]. Due to this, this so-called “slow life” appears to be out of thermodynamic equilibrium. More accurately, they are subject to additional rate-limiting steps that are presently not fully understood.

References:

Chadwick, G. L. et al. Comparative genomics reveals electron transfer and syntrophic mechanisms differentiating methanotrophic and methanogenic archaea. *PLoS Biol* 20(1): e3001508. (2022) <https://doi.org/10.1371/journal.pbio.3001508>

Murali, R. et al. Physiological potential and evolutionary trajectories of syntrophic sulfate-reducing bacterial partners of anaerobic methanotrophic archaea. *PLoS Biol* 21(9): e3002292. (2023) <https://doi.org/10.1371/journal.pbio.3002292>

Thauer, R. K. et al. Energy conservation in chemotrophic anaerobic bacteria. *Bacteriol Rev* 41(1): 100-80. <https://doi.org/10.1128/br.41.1.100-180.1977>

Dolfing, J. The microbial logic behind the prevalence of incomplete oxidation of organic compounds by acetogenic bacteria in methanogenic environments. *Microb Ecol* 41, 83–89 (2001). <https://doi.org/10.1007/s002480000076>

Galushko, A. & Kuever, J. *Desulfovibrio*†. In *Bergey's Manual of Systematics of Archaea and Bacteria* (eds M.E. Trujillo, S. Dedysh, P. DeVos, B. Hedlund, P. Kämpfer, F.A. Rainey and W.B. Whitman). (2019) <https://doi.org/10.1002/9781118960608.gbm01035.pub2>

L585: Figure 4 shows 3 ASVs of *Candidatus Frackibacter*, so it is more than one bacterium. There seems to be some diversity within *Candidatus Frackibacter*.

The three sequences were highly similar to each other (99.6% similarity); this information has been added at Line 264. It may be that these are multiple operons from the same bacterium or strain-level heterogeneity.

L592-594: Is the single, abundant *Sphingomonas* ASV indicative of a deep mine environment, but not a surface contaminant? There is value in that conclusion, but that doesn't come across.

In addition to our observations, the presence of *Sphingomonas echinoides* has been observed as a contaminant in mine environments as noted on Line 434. While this does seem to indicate that it may be indicative of mine environments, there is insufficient evidence to support this based on the limited amount of data available from other mine environments.

L596-603: Bringing in new lines of evidence in the conclusion section adds confusion, suggest revising to focus on these results.

We agree. The end of the final paragraph within the Discussion has now been rewritten to explore this in more detail, beginning at Line 628.

General Comments:

Paragraph beginning at L126: The motivation, as written, for this study seems to be new methods applied to unique samples. What was the hypothesis and motivation for designing experimental work and studying these samples?

The motivation of the study has been clarified and stated more clearly at Line 128.

Results Section: Several of the results paragraphs have discussion-like comparisons to other studies and sometimes the results aren't presented at all (i.e., L279-283). The reader needs to know what was measured in this study.

We have reviewed the results section and any writing beyond simple comparison of data has been moved to the discussion. The missing citation to the geochemical results have been added.

Discussion Section: The repetition through the first four discussion paragraphs without presenting new lines of evidence to support the contention of *Candidatus Frackibacter* being native to fracture water needs to be revised.

The beginning of the discussion section has been rewritten to reduce repetition and strengthen support for the conclusions.

Focus on two ASVs in Discussion: The Discussion largely focuses on two ASVs of the ~400 found. Given this deep dive into these two ASVs, their metabolism, global distribution, ecology, etc., it needs a different method. I hate suggesting "deeper sequencing" because I work in low biomass environments and low volume samples, but it would seem to get at your question more directly. If these two ASVs are so abundant, a metagenome should produce good metagenome assembled genomes. This would add some weight to those discussions surrounding the two ASVs and their metabolism (i.e., L552). Ultimately, this gets back to my first general comment,

what was the hypothesis or question that motivated this work? It appears to be studying the ecological role of “indicator” or abundant microbial species in these two habitats. 16S amplicon sequencing is not the appropriate method to get at this question. It is for a survey of the entire community, but that is not what is presented in this paper.

We acknowledge that the manuscript focuses on *Candidatus Frackibacter* and *Sphingomonas echinoides*. While the initial study was aimed at evaluating the scale and scope of the microbial profiles in the fracture networks, the subsequent focus has been placed on these two organisms due to their high relative abundance in the communities (sometimes as high as 99%). This information has also been added to the relevant Results sections. To reflect the wider ASV spectrum that was investigated, we have also included the names of other organisms identified in this study in the revised manuscript beginning at Line 601.

We agree that metagenomic sequencing could help clarify how *Candidatus Frackibacter* is sustained in this environment and hope to complete this analysis as a follow-up to this study.

Overall, the paper does a good job of demonstrating the 12299 fracture water and biofilm samples are not contaminated.

We appreciate the reviewer’s confirmation that this was effectively communicated.

Manuscript COMMSENV-24-0835-T

Dear Dr. D'Arcy Meyer-Dombard and Joe Aslin,

We appreciated the reviewers second evaluation of our manuscript and were pleased that the response to the edits we made was positive. We are very pleased that you are willing to publish this article pending the minor revisions requested.

We were pleased to see that Reviewer 1 and 2 had no further comments. We have responded to the minor points raised by Reviewer 3 as described in red text below. We do appreciate their work in helping us improve the manuscript.

We have resubmitted the manuscript with changes tracked and a changes accepted version. We have also gone through the editorial requests table and reviewed all formatting and policy requirements.

We greatly look forward to seeing this manuscript published in Nature Communications Earth and Environment.

Thank you for your work in overseeing this process.

Sincerely,

Greg Slater, on behalf of the authors

Rebuttal Letter

Deep terrestrial indigenous microbial community dominated by *Candidatus Frackibacter* by

Sian E. Ford¹, Greg F. Slater¹, Katja Engel², Oliver Warr^{3,4}, Garnet S. Lollar³, Allyson Brady^{1,5}, Josh D. Neufeld³, Barbara Sherwood Lollar^{2,6}

We were pleased to see that Reviewer 1 and 2 had no further comments. We have responded to the minor points raised by Reviewer 3 as described in red text below. We do appreciate their work in helping us improve the manuscript.

Reviewer #3 (Remarks to the Author):

The authors have made many changes that have improved the manuscript. Below are some specific comments.

Introduction Paragraph 2: The case for Mars is made well, but not the other satellites. Those should be removed because their case hasn't been made and they are ocean worlds. Does this come back around in the discussion?

Response: Please see the added sentence at Line 55. Kidd Creek has been specifically referenced by major NASA reports (Refs 36 and 37) as an analogue target for ocean world exploration of satellites like Enceladus and Europa. We have included a new reference here (Ref 38), the NASA Europa Mission planning report (Hand et al. 2016), that again makes specific reference to Kidd Creek for ocean world exploration and hence supports our sentence. We have removed the inclusion of Titan as it is the least relevant.

L155: Not statistically different, so $p > 0.01$, right? Depends on how you stated your hypothesis and set up the test. A few lines down waters are significantly different using the same test and p is still < 0.01 .

Response: We are grateful that the reviewer caught this error, which is now corrected at Line 157. Further clarification has been made at Line 157 as this is the comparison between FW12261 and FW12299 which were not statistically different in terms of geochemistry and stable isotopes. At Line 164 the comparison being made is pairwise between the service water and FW12261, and service water and FW12299. This sentence concludes that the service water is statistically different from either of the two fracture waters.

L158-161: Something is confusing here. A few sentences ago the waters from 12299 and 12261 were statistically identical and now one is more representative than the other. Please revise.

Response: This sentence has been removed because the representative nature of FW12299 is discussed in better context in the Discussion section beginning at Line 324, with specific mention at Line 390.

L237: ASVs are usually number ordered by their abundance in the batch of samples processed. If all these samples were processed together (FW and SW), then who are ASVs 1-13 because those are high abundance ASVs. Are you referencing the Supplemental ASV table at line 244? I don't understand what is presented in that file.

Response: In the QIIME2 pipeline as we use it, ASVs are not ordered by their abundance. Instead, their order in the ASV table is arbitrary. ASVs are not numbered in the QIIME2 pipeline, but we do assign a row identifier to our ASVs in the ASV table, which then gets assigned to each ASV in the visualizations.

Yes, line 244 is supposed to reference the Supplemental ASV table but was referencing the wrong table. The sentence now reads:

“*Thiobacillus sp.* (32.05 ± 0.55%) and members of the family *Comamonadaceae* (20.5 ± 1.0%) were the two most abundant ASVs in both SW samples (Figure 4, FORDSE_MS_ADDITIONAL_DATA.xlsx).

Results: A rank-order list of the top 20 most abundant ASVs and their relative abundances from the different samples would really help to contextualize how abundant these ASVs are relative to everything else that was found in these samples. Like Fig. 4, but on Fig. 4 it is not clear if these are the top 26 most abundant ASVs from the entire dataset (the ASV numbering would imply not).

Response: As explained the response to the reviewer question above, the ASV numbers are arbitrary and are not indicative of abundances. Figure 4 does indeed show the most abundant ASVs identified in those samples. We have modified the Figure caption to highlight this.

L283: I understand FAPROTAX is a published method, but the organism in question is a *Candidatus Frackibacter*. How well does FAPROTAX deal with uncultured and undescribed species?

Response: Although FAPROTAX would not perform well on completely novel lineages, the taxonomic resolution of *Candidatus Frackibacter* to the genus level is sufficient for FAPROTAX to assign functional groups by extrapolating knowledge from the related genus *Fuchsiella* from the same family *Halobacteroidaceae*. This was done because SILVA 138 assigned *Fuchsiella* to these

ASVs. Thus, FAPROTAX inferred functions are for the *Fuchsiella* genus, as we discuss in the manuscript, and additional metagenomic and cultivation-based approaches would be required to confirm putative functions for *Candidatus* Frackibacter ASVs we detected. For lack of specific recommendations, we have not made changes in response to this reviewer question.

L356: The comparison to food fermentation factories needs to be described better. What does this tell us about your study?

Response: We have adjusted this sentence (Line 366) to highlight that *Candidatus* Frackibacter has been identified elsewhere. Comparison to food fermentation is not relevant to the study so this specificity has been removed.

L495: “producing delivering” please revise.

Response: This error has been corrected, Line 510.

L510-522: I still find this discussion of the metabolic potential speculative.

Response: We have clarified the speculative nature of this discussion point regarding *Candidatus* Frackibacter’s metabolic potential at Line 531.

Conclusion section: Can you provide a 2-3 sentence conclusion of the stable isotope analysis? That would help tie up those data.

Response: We have increased the clarity of the stable isotope analysis by adding a statement at Line 468 contrasting the isotopic results between the two groups of samples to help tie up that argument. We have also included a short statement in the Conclusions section relating the patterns of stable isotope results, along with the Faprotax results, to the inferred metabolic activities.

Results/Figure 5. I don’t fully understand what is being correlate and how it matters

Response: Clarification on these results and the heatmap presented in Figure 5 have been made beginning at Line 242. The analysis being carried out here is using Spearman’s rank correlation to reveal how similar (or dissimilar) a sample is based on the taxa detected through 16S rRNA gene analysis. Here all the taxa detected for a give sample are treated as a profile or fingerprint which is representative of that sample.